# USE: Enhancing Mixed-Motive Cooperation via Unified Self and Collective Rewards

## Abstract

Mixed-motive cooperation requires agents to balance individual and collective rewards, often leading to tension between self-interest and cooperation. Conventional methods typically treat individual and collective rewards as completely independent, solving mixed-motive cooperation by maximizing their weighted sum (sometimes with auxiliary constraints). Because maximizing either the individual or the collective reward alone is sufficient to increase the weighted sum, such a design might lead to incorrect credit assignment and converge to overly selfish or altruistic policies. To address this, we propose a novel method named Unifying Self and collEctive rewards (USE). USE decomposes the individual reward into an independent part, unaffected by others, and a dependent part, shaped by interactions with others, then correlates the individual reward with the collective reward via the dependent part, as both the dependent part and collective reward arise from the interaction (including cooperation or defection) of agents. This coupling transforms maximizing individual and collective rewards into intrinsically correlated objectives, so that optimizing one implicitly promotes the other, reducing the risk of overly selfish or altruistic convergence. We conduct extensive experiments in mixed-motive cooperation tasks, demonstrating the effectiveness of USE. Interestingly, we find that the correlation between individual and collective rewards, to a certain extent, reflects the cooperative tendency of the agents. Our code is available at https://anonymous.4open.science/r/QPC-B6FD.

## 1 Introduction

Multi-agent cooperation is the process by which multiple agents coordinate their behaviors to maximize collective performance (Du et al., 2023; McKee et al., 2020). Existing multi-agent reinforcement learning (MARL) research focuses on pure-motive cooperation scenarios, where individual goals are naturally aligned with collective outcomes, making cooperation beneficial for all agents (Yu et al., 2022; Wang et al., 2020; Li et al., 2024a). However, real-world scenarios, such as traffic control (Kolat et al., 2023) and resource allocation (Mu et al., 2024), often involve mixed-motive cooperation, where environmental constraints compel agents to balance individual and collective rewards (i.e., selfishness and altruism). Purely selfish behavior undermines collective outcomes under such constraints and ultimately harms the long-term individual returns (Leibo et al., 2017). For example, as illustrated in Figure 1 (a) and (b), agents collect apples to gain rewards, but the apples (reward source) appear only when agents perform the unprofitable action of cleaning a polluted river (Hughes et al., 2018). In this case, agents must balance their incentive to maximize apple collection with the need to take turns cleaning the river. If all agents pursue only individual rewards (e.g., collecting apples), the polluted river will hinder apple growth and undermine the collective reward in the long term (Figure 1 (a)). Conversely, cleaning the river yields no individual reward. Therefore, mixed-motive cooperation settings are more challenging than pure-motive settings due to the fundamental misalignment between agents' short-term individual rewards and the long-term collective reward (Van Lange et al., 2013; Dawes, 1980; Kollock, 1998).

To address mixed-motive cooperation, prior methods typically optimize a weighted sum of individual and collective rewards, sometimes with additional constraints (Li et al., 2024b; Roesch et al., 2024; Wang et al., 2019; Jaques et al., 2019; Balduzzi et al., 2018; Yang et al., 2020). However, this formulation treats the two as separate objectives; since maximizing either component alone can increase the weighted sum, it often results in incorrect credit assignment and fosters free-riding or

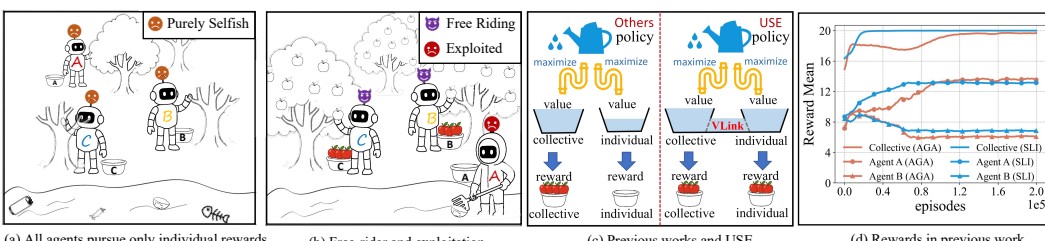

(a) All agents pursue only individual rewards     (b) Free-rider and exploitation     (c) Previous works and USE     (d) Rewards in previous work

Figure 1: Diagram of mixed-motive cooperation. (a) In Cleanup, no agent cleans the river, preventing apple regeneration and yielding no rewards. (b) One agent cleans but is exploited by others, leading to exploitation. (c) USE correlates individual and collective rewards via the dependent Q-value, enabling joint optimization. (d) In the Chicken Game, prior methods (e.g., AGA, SLI) exhibit exploitation, where one agent benefits disproportionately at the expense of others.

exploitation (Hughes et al., 2018; Wong et al., 2023). Consider the Cleanup environment in Figure 1 (a) and (b): after agent A cleans the river, the apple tree flourishes. Agents B and C benefit from the harvest without contributing, while B may mistakenly attribute the reward to its own actions, adopting free-rider behavior. Meanwhile, the gradient of A may be disproportionately amplified by collective rewards during training, leaving it exploited. A similar issue appears in the Chicken Game (Smith & Price, 1973) (Figure 1 (d)), where one agent consistently gains high rewards while the other receives little. Notably, the collective reward and one agent's payoff often increase as the other's decreases, suggesting convergence toward overly selfish or altruistic policies driven by collective incentives.

These observations highlight that treating individual and collective rewards as completely independent objectives may be inappropriate in mixed-motive settings, because it is easier to learn a policy that maximizes one reward rather than both. This runs counter to the mixed-motive setting, where agents are expected to balance individual reward with collective reward. A solution to facilitate the learning might be to explicitly correlate the two rewards so that they function as a unified objective: maximizing one implicitly promotes the other (see Figure 1 (c), right). Fortunately, individual reward can be naturally decomposed into (1) an independent reward gained from the agent's own action, and (2) a dependent reward gained via interactions with others (De Lange, 1980; Rusbult & Van Lange, 2003). Notably, the dependent reward is correlated with the collective reward, as both arise from interactive behaviors. This allows us to establish a correlation between individual and collective rewards, enabling agents to improve collective rewards while pursuing individual goals.

Inspired by that, we propose a method called Unifying Self and collEctive rewards (USE), which addresses the problem by unifying the pursuit of individual and collective rewards (i.e., Q-value). Specifically, USE consists of two modules: Individual Dependence Decomposition (IDD) and Individual Collective Coupling (ICC). The IDD models the individual Q-value from independent and dependent components. The ICC further correlates individual and collective Q-values by capturing the correlation (denoted as VLink) between the collective Q-value and the dependent component. By optimizing both the individual Q-value and its alignment with the collective Q-value, agents can maximize their individual rewards while preserving collective reward, thereby mitigating the issue of overly selfish or altruistic policies. We conducted extensive experiments across various environments, and the results demonstrate the effectiveness of USE. Interestingly, we also found that the correlation between the individual and collective, to some extent, reflects the agents' cooperative tendency.

In summary, our contributions are: (i) we propose a method called USE to address the mixed-motive cooperation; (ii) To the best of our knowledge, USE is the first work to leverage the correlation between individual and collective rewards to facilitate a balance between them; (iii) Experiments on various environments illustrate the effectiveness of USE, and we found that the correlation between individual and collective rewards reflects the cooperative tendency of agents.

## 2 RELATED WORK

In mixed-motive cooperation, individual and collective incentives are not perfectly aligned (Leibo et al., 2017). Excessive selfishness reduces collective reward, whereas excessive altruism risks exploitation (Rapoport, 1974). Some works rely on reward shaping and weighting. For example, Yu et al. (2022) directly uses the collective reward as the learning signal, and Peysakhovich & Lerer (2017) mixes individual rewards with the average collective reward. Wang et al. (2019) evolves the weight

between short-term self-interest and long-term collective reward, while Roesch et al. (2024) attempt to measure the selfishness level in matrix games to assign precise weights to the collective reward. However, such heuristics have difficulty adapting to complex environments. Several works introduce auxiliary constraints. Jaques et al. (2019) models teammates' behaviors to capture social influence constraints. And other works incorporate benevolence (Tennant et al., 2024), morality (Perolat et al., 2017), or reputation (Anastassacos et al., 2021) as constraints. Inequity aversion (Hughes et al., 2018) has also been extended to Markov games by penalizing payoff inequality. Yang et al. (2020); Li et al. (2024b) align individual updates with the collective objective by modifying policy gradients. Although these methods achieve partial success, they still largely treat individual and collective rewards as separate objectives, remaining vulnerable to free-riding or exploitation.

## 3 PRELIMINARY

In mixed-motive cooperation tasks, the structural constraints of the environment compel agents to strike a balance between selfishness and altruism so as to maximize their long-term returns. Formally, the mixed-motive cooperation can be formulated as a partially observable general-sum Markov game with $\mathcal{N}$ players: $\langle \mathcal{N}, \mathcal{A}, \mathcal{S}, \mathcal{P}, \mathcal{O}, \Omega, \mathcal{R}, \gamma \rangle$, where $\mathcal{N} = \{1, ..., n\}$ is a set of agents, $\mathcal{A}$ is the set of joint actions, $\boldsymbol{a}_t = \{a_t^i\}_{i \in \mathcal{N}}$, where $a_t^i$ denotes the action of agent $i$ at step $t$. $\mathcal{S}$ is a set of global states $\boldsymbol{s}_t$, which evolve according to the transition function $\mathcal{P}(\boldsymbol{s}_{t+1}|\boldsymbol{s}_t, \boldsymbol{a}_t)$. The global state $\boldsymbol{s}_t$ at time step $t$ is usually a high-dimensional vector that concatenates the states of each agent and the relevant environment. $\mathcal{O}$ is a set of local observations $\boldsymbol{o}_t$ received by agents, and the local observation of agent $i$ is $o_t^i \sim \Omega(\boldsymbol{a}_t, \boldsymbol{s}_t)$, where $\Omega$ is the observation mapping function. $\mathcal{R} = \{r_t^c, \{r_t^i\}_{i \in \mathcal{N}}\}$. Here, $r_t^i$ denotes the individual reward at step $t$. $r_t^c$ denotes the collective reward (or social welfare) at step $t$, which is typically defined as the sum of all individual rewards. $\gamma \in [0, 1)$ is the discount factor. In mixed-motive settings, the **learning objective** of each agent is the balance between maximizing their individual rewards and the collective reward (Du et al., 2023; Leibo et al., 2017).

## 4 METHOD

To jointly optimize individual and collective rewards, we propose a method called Unifying Self and collEctive rewards (USE), which adopts the centralized training with decentralized execution (CTDE) framework. Different from previous works, which treat individual reward and collective reward as independent objectives, USE explicitly couples the expected cumulative values of individual and collective rewards (*i.e.*, Q-value) to unify them for the joint optimization. Specifically, as illustrated in Figure 2, USE has two modules to address the issue: Individual Dependence Decomposition (IDD) and Individual Collective Coupling (ICC). IDD models the individual Q-value as the combination of the independent Q-value gained from an agent's own actions and dependent Q-value gained from interactions with other agents. ICC unifies the pursuit of individual and collective rewards by leveraging the value-level alignment between the dependent Q-value and the collective Q-value, thereby facilitating their joint optimization and mitigating the free-riding and exploitation issues encountered in previous works. In the following, we first present IDD and ICC, then describe how to optimize USE based on the results of IDD and ICC.

### 4.1 INDIVIDUAL DEPENDENCE DECOMPOSITION (IDD)

From the perspective of reward attribution, the individual reward of an agent can be grouped into (1) independent reward, the reward gained from its own actions; (2) dependent reward, the reward gained from interactions with other agents. Considering the example in Introduction again (Figure 1 (a) and (b)), an agent receives a reward for collecting apples. This reward originates not only from the act of picking apples itself, but also from prior cooperation with other agents to clean the river, which enabled the apple trees to bear fruit. Therefore, we estimate the Q-value of an agent from the perspectives of independence and dependence in the Individual Dependence Decomposition (IDD). As illustrated in Figure 2 (a), when agent $i$ takes action $a_t^i$ ($a_t^i \sim \pi^i(a_t^i|o_t^i, \tau_t^i)$ and $\tau_t^i = (o_1^i, a_1^i), ..., (o_{t-1}^i, a_{t-1}^i)$) in response to the environment[1], we use $a_t^i$, the observation $o_t^i$, and the global state $\boldsymbol{s}_t$ as inputs. Then, we model both the independent and dependent Q-value for agent $i$, calculating the individual Q-value.

---

[1] $a_t^i$ is obtained by feeding the concatenation of the observation ($o_t^i$) and trajectory to a recurrent neural network (RNN) (Graves, 2012). Focusing on joint optimization of individual and collective rewards, we omit further decision details.

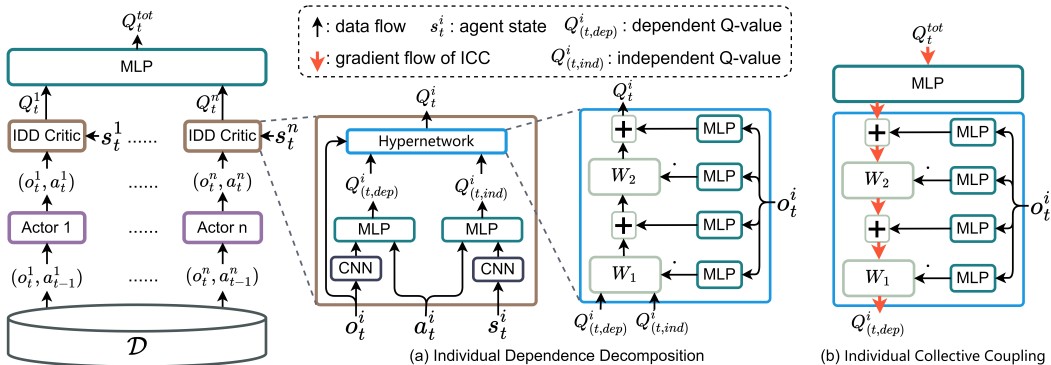

Figure 2: The framework of USE. (a) forms the Individual Dependence Decomposition (IDD). In IDD, the dependent Q-value and independent Q-value are fused via a hypernetwork to form the individual Q-value, where the hypernetwork is generated by the agent's observation. (b) illustrates the Individual Collective Coupling (ICC). In ICC, the agent's cooperative tendency is estimated by computing the partial derivative of the collective Q-value with respect to the dependent Q-value.

**Independent Q-value Estimation:** To estimate the independent Q-value of agent $i$, we remove the information of other agents from global state $s_t$, and denote the remaining as the state of agent $i$:

$$s_t^i = \text{Prune}(s_t), \tag{1}$$

where $\text{Prune}(.)$ denotes the removal operation. For row-vector state inputs, we remove dimensions corresponding to other agents, retaining only information related to agent $i$ and the environment (details are provided in Appendix B.1.1). We then estimate the independent Q-value by:

$$Q_{(t,\text{ind})}^i = f_{\text{ind}}(\left[\text{Enc}(s_t^i), a_t^i\right]), \tag{2}$$

where $\text{Enc}(.)$ denotes the encoder of agent $i$'s state, implemented with a Convolutional Neural Network (CNN) (Krizhevsky et al., 2012). $[,]$ denotes concatenation. $f_{\text{ind}}$ is the mapping function realized with a Multi-Layer Perceptron (MLP). $Q_{(t,\text{ind})}^i$ denotes the independent Q-value.

**Dependent Q-value Estimation:** Empirical studies (Foerster et al., 2016) show that agents can typically interact only within a finite radius. In many environments (e.g., Cleanup), the exact boundaries of this "interaction neighborhood" are difficult to quantify. As a pragmatic compromise, we adopt the local observation window as an approximation of the interaction neighborhood. Thereby, we estimate the dependent Q-value with the observation ($o_t^i$) and the action ($a_t^i$) of agent $i$:

$$Q_{(t,\text{dep})}^i = f_{\text{dep}}(\left[\text{Enc}(o_t^i), a_t^i\right]). \tag{3}$$

$f_{\text{dep}}$ is the mapping function. We implement it with an MLP. $Q_{(t,\text{dep})}^i$ denotes the dependent Q-value.

**Q-value Estimation:** Although we model the individual Q-value from dependent and independent Q-values, their relationship cannot be expressed as a simple linear sum. Instead, we adopt a hypernetwork (Ha et al., 2017) to fuse the dependent and independent Q-values. Specifically, as illustrated in Figure 2 (a), we feed the observation $o_t^i$ into mapping functions to obtain the parameters for fusing them:

$$\{W_1^i, b_1^i, W_2^i, b_2^i\} = H(o_t^i), \tag{4}$$

where $H(o_t^i)$ denotes the hypernetwork that generates the parameters $W_1^i = f_{W1}(o_t^i)$, $b_1^i = f_{b1}(o_t^i)$, $W_2^i = f_{W2}(o_t^i)$, and $b_2^i = f_{b2}(o_t^i)$. We implement each function $f_{W1}, f_{W2}, f_{b1}, f_{b2}$ with an MLP. Next, we estimate the individual Q-value of agent $i$ $Q_t^i$ with:

$$Q_t^i = f_i(Q_{(t,\text{ind})}^i, Q_{(t,\text{dep})}^i) = W_2^i \left( W_1^i \left[ Q_{(t,\text{ind})}^i, Q_{(t,\text{dep})}^i \right] + b_1^i \right) + b_2^i. \tag{5}$$

## 4.2 INDIVIDUAL COLLECTIVE COUPLING (ICC)

To jointly maximize the individual and collective rewards, previous methods treat individual and collective rewards as independent objectives, either maximizing their weighted sum or introducing auxiliary constraints to balance the two. However, as we discussed previously, these solutions might

lead agents to converge to either overly selfish or overly altruistic policies, resulting in free-riding or being exploited. The Individual Collective Coupling (ICC) aims to correlate the individual reward with the collective reward, enabling their unification to facilitate more effective joint optimization.

Specifically, we first estimate the collective (global) Q-value using individual Q-values of all agents:

$$Q_t^{\text{tot}} = f\left[Q_t^1, ... Q_t^n\right], \tag{6}$$

where $f$ denotes the mapping function, which is implemented using an MLP, and $Q_t^{\text{tot}}$ denotes the collective Q-value. It is worth clarifying that the estimation of $Q_t^{\text{tot}}$ is not required during execution; therefore, Eq. 6 does not introduce any information leakage among agents during testing.

Considering that both dependent and collective rewards rely on agent interactions in the group, we can establish a correlation between individual and collective rewards via the dependent reward. To achieve this, we first analyze the relationship between dependent and collective Q-values. When an agent takes an action increasing its dependent reward (*i.e.,* gaining more rewards through interactions with others), its dependent Q-value rises correspondingly, leading to two collective Q-value changes: (1) the action benefits the group, increasing the collective Q-value (Figure 3 (a)); or (2) the action harms the group, decreasing the collective Q-value (Figure 3 (b)). Based on this observation, we propose to quantify an agent's group contribution by evaluating how agent $i$'s dependent Q-value-increasing behaviors affect the collective Q-value:

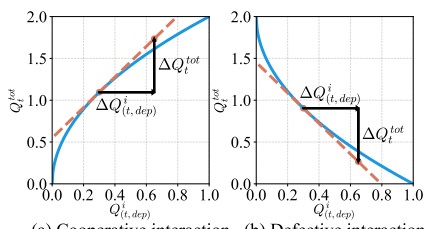

(a) Cooperative interaction  (b) Defective interaction

Figure 3: $Q_t^{\text{tot}}$ and $Q_{(t,\text{dep})}^i$. The derivative $g_t^i$ (VLink) is used to characterize cooperative and defective interactions.

$$g_t^i = \lim_{\Delta Q_{(t,\text{dep})}^i \to 0} \left(\Delta Q_t^{\text{tot}} / \Delta Q_{(t,\text{dep})}^i\right) = \left(\partial Q_t^{\text{tot}} / \partial Q_{(t,\text{dep})}^i\right). \tag{7}$$

where $\Delta Q_{(t,\text{dep})}^i$ denotes the increment of dependent Q-value of agent $i$, and

$$\Delta Q_t^{\text{tot}} = f\left[Q_t^1, ..., Q_t'^i, ..., Q_t^n\right] - f\left[Q_t^1, ..., Q_t^i, ..., Q_t^n\right], \tag{8}$$

$$Q_t'^i = f_i\left(Q_{(t,\text{ind})}^i, Q_{(t,\text{dep})}^i + \Delta Q_{(t,\text{dep})}^i\right). \tag{9}$$

When the agent $i$ provides beneficial contribution to the group, *i.e.,* the action of agent $i$ is beneficial for the collective reward, $g_t^i$ is positive. In contrast, when agent $i$ provides harmful contribution to the group, $g_t^i$ is negative. In this way, maximizing $g_t^i$ encourages policies that simultaneously improve both the agent's individual reward and the collective reward, thus avoiding convergence to overly selfish or overly altruistic behaviors. Since $g_t^i$ unifies the individual and collective rewards with the dependent Q-value, we refer to it as the **Value Link (VLink)**.

### 4.3 MODEL LEARNING

#### 4.3.1 CRITIC LEARNING

To accurately estimate the individual Q-value and collective Q-value, we employ two TD errors (Peng & Williams, 1993; Mnih et al., 2015; Rashid et al., 2020) to optimize the estimation:

$$\mathcal{L}_i = \mathbb{E}_{\mathcal{D}}\left[\left((r_t^i + \gamma \bar{Q}_{t+1}^i) - Q_t^i\right)^2\right], \tag{10}$$

$$\mathcal{L}_t = \mathbb{E}_{\mathcal{D}}\left[\left((r_t^c + \gamma \bar{Q}_{t+1}^{\text{tot}}) - Q_t^{\text{tot}}\right)^2\right], \tag{11}$$

where $\bar{Q}_t^i$ and $\bar{Q}_t^{\text{tot}}$ are target values whose parameters are periodically copied from the online networks and kept fixed for a number of iterations. Therefore, the learning target of critic is:

$$\mathcal{L}_c = \sum_{i=1}^n \mathcal{L}_i + \mathcal{L}_t. \tag{12}$$

#### 4.3.2 ACTOR LEARNING

The action advantage for agent $i$ is computed by:

$$A_t^i = Q_i(o_t^i, \tau_t^i, a_t^i) - \sum_{a \in \mathcal{A}_i} \pi_i(a \mid o_t^i, \tau_t^i)\, Q_i(o_t^i, \tau_t^i, a), \tag{13}$$

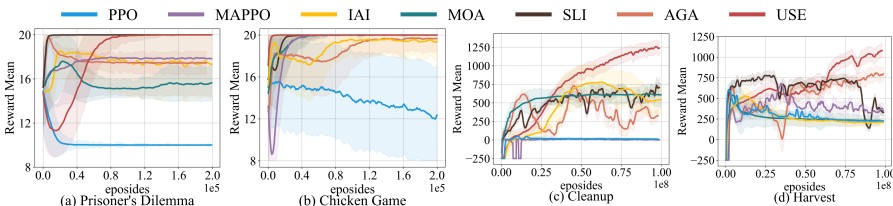

Figure 4: The benchmark results of collective reward: the x-axis denotes the training episodes, and the y-axis indicates the average collective reward. USE achieves the best result in all environments.

where $A_t^i$ represents the advantage function of agent $i$. $\pi^i$ denotes the policy of agent $i$. $\tau_t^i$ denotes the historical trajectory at step $t$. $\mathcal{A}_i$ represents its action space.

To jointly maximize the individual and collective Q-values, we optimize the policy of agent $i$ as:

$$\nabla_{\theta^i} J(\theta^i) = \nabla_{\theta^i} \log \pi^i(a_t^i | o_t^i, \tau_t^i) \cdot \left[ A_t^i + \lambda g_t^i \right], \tag{14}$$

where $\theta_i$ represents the parameters of agent $i$'s actor, and $\lambda$ is a hyperparameter (we set $\lambda = 1$ in all cases). During actor training, the advantage and VLink are treated as constants (i.e., gradients are stopped and do not backpropagate through them). The proof of convergence is presented in Appendix C.1, and further details of the method are illustrated in Appendix C.3 and Algorithm 1.

## 5 EXPERIMENT

We conduct extensive experiments to evaluate the performance of USE, and particularly focus on the research questions: i) Is USE effective in mixed-motive cooperation tasks (**RQ1**)? ii) Can USE generalize to more challenging environments (**RQ2**)? iii) What are the key factors supporting the performance of USE (**RQ3**)? iv) Does USE converge to extremely selfish or altruistic policies (**RQ4**)? v) Does the VLink reflect agents' contributions to the collective reward (**RQ5**)? vi) Can Individual Dependence Decomposition (IDD) effectively capture dependent and independent values (**RQ6**)?

### 5.1 EXPERIMENT SETTINGS

To evaluate the effectiveness of USE, we assessed it across 5 environments: Prisoner's Dilemma, Chicken Game, Cleanup, Harvest, and Allelopathic Harvest (Rapoport, 1974; Smith & Price, 1973; Hughes et al., 2018; Agapiou et al., 2022). These environments include matrix games for one-shot mixed-motive interactions, sequential settings for long-term shared dynamics, and adversarial interactions (details are in Appendices A). We selected 6 representative methods from top-tier conferences as baselines[2], including: PPO (Schulman et al., 2017) (purely selfish, OpenAI 2017), MAPPO (Yu et al., 2022) (purely cooperative, NeurIPS), IAI (Hughes et al., 2018) (inequality aversion, NeurIPS), MOA (Jaques et al., 2019) (social influence, ICML), SLI (Roesch et al., 2024) (reward reweighting, AAMAS), and AGA (Li et al., 2024b) (gradient adjustment, NeurIPS). The baselines cover various strategies for optimizing both individual and collective rewards. For a fair comparison, we use official implementations or recommendations from the original papers, ensuring each baseline operates under its optimal configuration. Each experiment was repeated 5 times with random seeds, and results are reported as mean ± standard deviation. Implementation details and hyperparameters are in Appendix C.4.

### 5.2 BENCHMARK RESULTS

We compare USE to baselines during online evaluation, and the results are illustrated in Figure 4. Since the collective reward is defined as the sum of all individual rewards in the evaluation environments, the trends of individual rewards are aggregated into the collective reward, and Figure 4 reflects both individual and collective rewards in mixed-motive settings simultaneously.

From Figure 4, we can observe that: (1) Compared to other methods, USE achieves the best performance, particularly in environments requiring long-term interactions (Cleanup and Harvest).

---

[2]QMIX, a primary multi-agent reinforcement learning method, is designed mainly for pure cooperative rather than mixed-motive tasks. Thus, our comparison results with QMIX are in Appendix B.3.

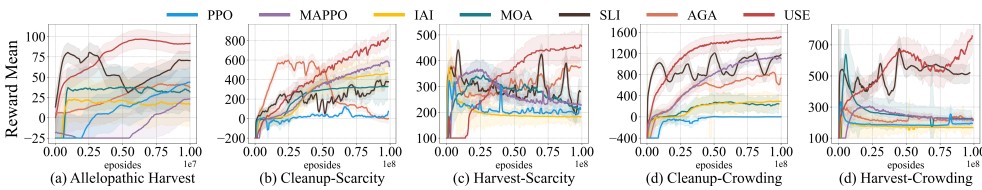

Figure 5: The results of more challenging environments.

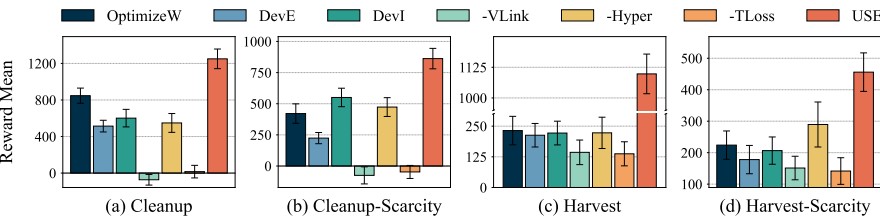

Figure 6: The Ablation Study of USE. We tested the average collective reward under different variants of USE. For clarity and comparison, the y-axis in (c) is truncated to the range [300, 950].

These results demonstrate the effectiveness of USE in mixed-motive cooperation (RQ1). (2) The advantage of USE is less pronounced in the Prisoner's Dilemma and Chicken Game compared to Cleanup and Harvest. This is likely because the former are one-shot matrix games with lower complexity, where most methods achieve satisfactory results. In contrast, Cleanup and Harvest require long-term interactions and involve more agents, making them inherently more challenging. As a result, the superior performance of USE is more evident in these complex environments. (3) We observe that USE exhibits slower improvement during the early training phase, performing worse than the baseline methods. This may be due to incomplete convergence of Q-value estimates in the early phase (warm-up), which limits the joint optimization of individual and collective rewards.

## 5.3 MORE CHALLENGING RESULTS

To further validate USE, we evaluated it in more complex environments, including Allelopathic Harvest and two more challenging variants of Cleanup and Harvest (Scarcity and Crowding):

- **Allelopathic Harvest:** The environment that introduces adversarial interactions among agents in addition to selfish and altruistic behaviors;
- **Scarcity:** The apple regeneration rate is reduced to increase competition. We denote the corresponding environment as **Cleanup-Scarcity** and **Harvest-Scarcity**;
- **Crowding:** The number of agents is increased to 10 to induce complex interactions. We denote the corresponding environment as **Cleanup-Crowding** and **Harvest-Crowding**.

The results are presented in Figure 5. We can observe that USE achieves the best performance even in more challenging environments. This demonstrates the strong stability and generalization capability of USE in these settings (RQ2). We further validated the robustness and hyperparameter sensitivity of USE in Appendix B.2 and Appendix B.4, respectively.

## 5.4 ABLATION STUDY

To further investigate the factors that support the performance of USE (RQ3), we conduct an ablation study. Specifically, we compare the performance of USE with the following variants:

- **OptimizeW** optimizes the weighted sum of individual and collective rewards instead of optimizing with Eq. 14, verifying the necessity of correlating the individual and collective rewards.
- **DevE** correlates individual and collective rewards via the independent Q-value (*i.e.*, $g_t^i = \frac{\partial Q_t^{\text{tot}}}{\partial Q_{(t,\text{ind})}^i}$).
- **DevI** correlates individual and collective rewards via the individual Q-value (*i.e.*, $g_t^i = \frac{\partial Q_t^{\text{tot}}}{\partial Q_t^i}$).
- **-VLink** removes the term $g_t^i$ from the optimization of the agent (Eq. 14), which aims to validate whether unifying the individual and collective rewards is beneficial for jointly optimizing them.
- **-Hyper** replaces the hypernetwork used in the Q-value estimation in Section 4.1 with an MLP, aiming to verify the necessity of applying a hypernetwork.

Table 1: The Gini Coefficient of individual rewards in various methods. The optimal and suboptimal results are marked in bold and underlined respectively.

| Metric | PPO | MAPPO | IAI | MOA | SLI | AGA | USE |
|--------|-----|-------|-----|-----|-----|-----|-----|
| PD | $0.0\pm0.0$ | $0.06\pm0.03$ | $0.04\pm0.02$ | $0.42\pm0.28$ | $0.0\pm0.0$ | $0.02\pm0.01$ | **$0.0\pm0.0$** |
| CG | $\underline{0.03\pm0.02}$ | $0.22\pm0.02$ | $\underline{0.02\pm0.01}$ | $0.19\pm0.02$ | $0.16\pm0.04$ | $0.20\pm0.01$ | **$0.01\pm0.01$** |
| Cleanup | $\underline{0.06\pm0.02}$ | $0.21\pm0.15$ | $0.07\pm0.03$ | $0.20\pm0.04$ | $0.17\pm0.03$ | $0.19\pm0.11$ | **$0.02\pm0.01$** |
| Harvest | $\underline{0.01\pm0.01}$ | $0.06\pm0.01$ | $\underline{0.01\pm0.01}$ | $0.05\pm0.01$ | $0.03\pm0.01$ | $0.03\pm0.01$ | **$0.01\pm0.01$** |

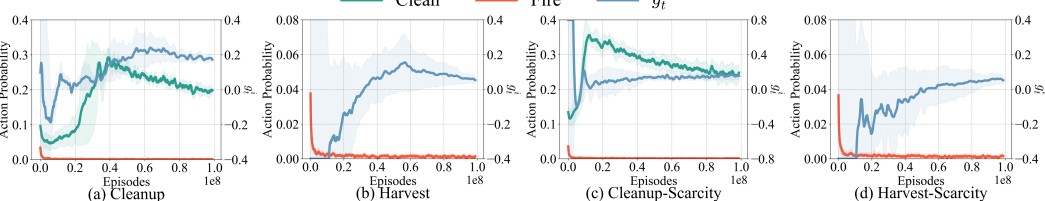

Figure 7: The curves of $g_t^i$ with the probability of clean and fire in Cleanup and Harvest. The x-axis represents the number of training steps. The left y-axis denotes probabilities of corresponding actions, while the right y-axis represents the Value Link ($g_t^i$).

- **-TLoss** removes the constraint $\mathcal{L}_t$ (Eq.11), validating the need for collective Q-value approximation.

The results are illustrated in Figure 6, from which we draw the following findings: (1) All variants of VLink (OptimizeW, DevE, DevI, and -VLink) reduce USE's performance to some extent. Among them, -VLink causes the most significant performance drop. This is because removing the constraint of $g_t^i$ degenerates the algorithm into a purely rational policy, severely weakening agents' cooperative tendencies and ultimately harming collective reward. As for other variants (OptimizeW, DevE and DevI), since cooperation and defection manifest through interaction (De Lange, 1980; Rusbult & Van Lange, 2003), these variants struggle to capture such dynamics effectively, leading to the decreased collective performance. However, as they still retain certain mechanisms that promote cooperation from different perspectives, they outperform the purely rational -VLink. (2) -Hyper decreases the performance in all cases. That implies the relationship among individual Q-value, independent Q-value, and dependent Q-value is far more complex, beyond what a simple MLP can capture. (3) -TLoss significantly degrades the performance. This is because $\mathcal{L}_t$ ensures accurate estimation of each agent's contribution to the group. Without the constraint imposed by $\mathcal{L}_t$, the model cannot effectively guide inter-agent cooperation.

### 5.5 FAIRNESS ANALYSIS

To further investigate whether USE–driven agents tend to converge to extremely selfish or altruistic policies, we conduct a fairness analysis using the Gini coefficient. The Gini coefficient is computed by $G = \frac{1}{2n^2\mu_t} \sum_{i=1}^n \sum_{j=1}^n |r_t^i - r_t^j|$, where $\mu_t$ denotes the mean value of all agents' rewards, and $n$ denotes the number of agents. Obviously, a lower Gini coefficient indicates smaller reward disparities among agents and less pronounced selfish or altruistic tendencies, reflecting a higher level of fairness.

The results are shown in Table 1. We observe that: (1) Compared with baselines, such as MAPPO, SLI, and AGA, USE achieves relatively lower Gini coefficients. This is because baselines treat the individual and collective rewards as completely independent objectives, making it difficult to jointly optimize both objectives, leading to the extreme selfish or altruistic behavior. In contrast, USE unifies the pursuit of the individual and collective rewards, effectively mitigating such issues. (2) PPO also exhibits low Gini coefficients. This is because PPO solely optimizes individual rewards without considering collective benefits. Due to the absence of collective reward optimization, all agents exhibit poor performance, resulting in smaller reward disparities, as shown in Figure 4 and Figure 5. In contrast, USE maintains low Gini coefficients while achieving the best overall performance.

### 5.6 VISUALIZATION OF VALUE LINK

To investigate whether the Value Link ($g_t^i$) provides a meaningful signal regarding the contribution of agent $i$'s actions, we visualize the curve of $g_t^i$ alongside the probabilities of the clean and fire actions

throughout the training process in the Cleanup and Harvest environments. In these environments, the clean action reflects cooperative behavior beneficial to the collective reward, while the fire action corresponds to defecting behavior that may harm the collective reward. The results are shown in Figure 7. We observe three interesting phenomena: (1) The trend of $g_t^i$ generally aligns with the probability of the clean action and is inversely correlated with that of the fire action. Specifically, $g_t^i$ tends to be positive when the clean probability is high or fire is low, and negative when clean probability is low or fire is high. This observation supports our hypothesis in Section 4.2, indicating that $g_t^i$ can, to some extent, reflect the agent's cooperative tendency (RQ5). (2) Across all cases, $g_t^i$ gradually converges to a positive value. Meanwhile, the fire probability decreases toward zero, while the clean probability increases and stabilizes. These trends reflect the effectiveness of USE in shaping cooperative, group-oriented behavior. (3) The absolute value of $g_t^i$ is initially large but gradually converges as training progresses. This is because $g_t^i$ is computed as the derivative of the collective Q-value with respect to the dependent Q-value. At the beginning of training, the policy parameters are randomly initialized, causing the neural network's outputs to have high variance.

### 5.7 Visualization of Dependent and Independent Q-values

To investigate whether Individual Dependence Decomposition (IDD) can effectively capture both dependent and independent values, we visualized the independent and dependent values for each action. Specifically, we randomly sampled 32 trajectories, each consisting of 1000 timesteps with 5 agents. At each timestep, every agent had 9 available actions. For each action, we computed the mean and standard deviation across all agents, timesteps, and trajectories, thereby estimating the distributions of both dependent and independent values associated with each action. Intuitively, since fire and clean inherently involve strong agent interactions, their dependent values are more pronounced. In contrast, during independent behaviors, fire is meaningless, so its independent value tends to be relatively low. The outcome is presented in Figure 8.

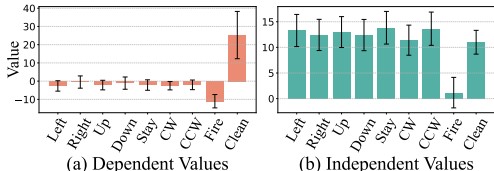

(a) Dependent Values    (b) Independent Values

Figure 8: To visualize the dependent and independent value functions, we consider the Cleanup environment, where each agent has nine discrete actions: moving left, right, up, down, staying, rotating clockwise (CW), rotating counterclockwise (CCW), firing, and cleaning. Figures (a) and (b) show the dependent and independent values of the 9 actions captured by IDD, respectively.

From the figure, we observe the following: (1) In Figure 8 (a), the dependent value of firing is negative because it corresponds to a defection strategy that undermines the collective reward, whereas the dependent value of cleaning is clearly positive as it represents a cooperative strategy that benefits the group. In Figure 8 (b), the independent value of firing is close to zero since firing offers no direct gain when the agent acts alone, while the independent value of cleaning remains positive because it still contributes to improving the individual reward. (2) in Figure 8 (a), the six movement and rotation actions show dependent values close to zero, reflecting that these actions do not involve interactions with others. By contrast, in Figure 8 (b), their independent values are positive, as these actions allow the agent to approach apples and thereby increase the chance of receiving rewards. This visualization demonstrates that IDD can effectively capture both the dependent and independent values associated with agents' actions (RQ6).

## 6 Conclusion and Limitations

In this paper, we propose a novel method called Unifying Self and collEctive rewards (USE) to address the issue of mixed-motive cooperation. Unlike previous methods that treat individual and collective Q-values as independent objectives, USE decomposes the individual Q-value into independent and dependent components, then correlates it with the collective Q-value via the correlation between the dependent component and the collective Q-value. It enables joint optimization of individual and collective objectives, reducing the likelihood of agents learning overly selfish or altruistic policies. Extensive experiments demonstrate the effectiveness of USE. Additionally, we find that the correlation between the dependent component and collective reward reflects the cooperative tendency of agents.

However, estimating an agent's dependent Q-value solely from local observations may still lack precision. Future research should explore more effective methods and frameworks for this estimation, which could in turn enable a more precise assessment of agents' cooperative tendencies.

## ETHICS STATEMENT

This work does not involve human subjects, personally identifiable information, or sensitive datasets. All experiments are conducted on publicly available multi-agent reinforcement learning environments. While advances in multi-agent learning may have potential downstream applications in strategic interactions or automated decision-making, our study is limited to simulated environments for scientific research purposes only. We have made efforts to ensure transparency and reproducibility by releasing our code and documenting experimental settings. We affirm that our work adheres to the ICLR Code of Ethics.

## REPRODUCIBILITY STATEMENT

We have made significant efforts to ensure the reproducibility of our work. The environments used in our experiments are publicly available, and detailed descriptions are provided in Appendix A. Additional ablations and sensitivity analyses are presented in Appendix B to validate the robustness of our findings. Implementation details of USE, including network architectures, hyperparameters, and training schedules, are provided in Section 4 and Appendix C, which also contains the complete proofs of the theoretical results. Our code is released through an anonymous supplementary link: `https://anonymous.4open.science/r/QPC-B6FD`. Together, these resources should allow readers to fully reproduce our results.

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

## A  ENVIRONMENT INTRODUCTION

### A.1  PRISONER'S DILEMMA AND CHICKEN GAME

We employ the Prisoner's Dilemma (PD) and the Chicken Game (CG) as experimental environments for mixed-motive cooperation. Their payoff matrices are shown in Table 2 (a) and Table 2 (b), respectively.

Table 2: Payoff matrices for Prisoner's Dilemma and Chicken Game.

| PD | C | D |
|----|---|---|
| C | (10,10) | (0,15) |
| D | (15,0) | (5,5) |

(a)

| CG | C | D |
|----|---|---|
| C | (10,10) | (5,15) |
| D | (15,5) | (0,0) |

(b)

## A.2 CLEANUP AND HARVEST

### A.2.1 CLEANUP

In the *Cleanup* game, the aim is to collect apples from a field. Each apple provides a reward of 1. The spawning of apples is controlled by a geographically separate aquifer that supplies water and nutrients. Over time, this aquifer fills up with waste, lowering the respawn rate of apples. For sufficiently high waste levels, no apples can spawn. At the start of each episode, the environment resets with waste just beyond this saturation point. To cause apples to spawn, agents must clean some of the waste.

Here we have a dilemma. Provided that some agents contribute to the public good by cleaning up the aquifer, it is individually more rewarding to stay in the apple field. However, if all players defect, then no one gets any reward. A successful group must balance the temptation to free-ride with the provision of the public good. Figure 9 (a) illustrates the state of the Cleanup environment, while Figure 9 (c) shows the observation of the orange agent.

### A.2.2 HARVEST

The goal of the *Harvest* game is to collect apples. Each apple provides a reward of 1. The apple regrowth rate varies across the map, dependent on the spatial configuration of uncollected apples: the more nearby apples, the higher the local regrowth rate. If all apples in a local area are harvested then none ever grow back. After 1000 steps the episode ends, at which point the game resets to an initial state.

The dilemma is as follows. The short-term interests of each individual lead toward harvesting as rapidly as possible. However, the long-term interests of the group as a whole are advanced if individuals refrain from doing so, especially when many agents are in the same local region. Such situations are precarious because the more harvesting agents there are, the greater the chance of permanently depleting the local resources. Cooperators must abstain from a personal benefit for the good of the group. Figure 9 (b) illustrates the state of the Harvest environment, while Figure 9 (d) shows the observation of the orange agent.

### A.2.3 SCARCITY AND CROWDING

For the Cleanup and Harvest, we adopt the default hyperparameter settings from Hughes et al. (2018). Further, we reduce the apple spawning frequency by half to encourage competition, yielding Cleanup-Scarcity and Harvest-Scarcity. Additionally, we increase the number of agents to 10 to enhance environmental complexity, resulting in action spaces of $9^{10}$ and $8^{10}$, denoted as Cleanup-Crowding and Harvest-Crowding. Detailed hyperparameter settings for the environment can be found in Table 3.

## A.3 ALLELOPATHIC HARVEST

The Allelopathic Harvest (Agapiou et al., 2022) is one of the most complex mixed-motive cooperation scenarios in the MeltingPot environment, as it not only involves the tension between selfish and altruistic behaviors but also incorporates adversarial interactions among teams. Featuring a larger state and action space as well as the presence of adversarial agents, it provides a particularly challenging setting for evaluating our method. The core task involves agents planting and harvesting different types of berries to maximize individual rewards. The environment consists of multiple "berry patches" where agents can plant and harvest red, green, and blue berries. Agents have heterogeneous color preferences (a red-preferring agent receives a reward of 2 for harvesting red berries and 1 for harvesting berries of other colors). Berries grow over time and can be harvested once ripe. The

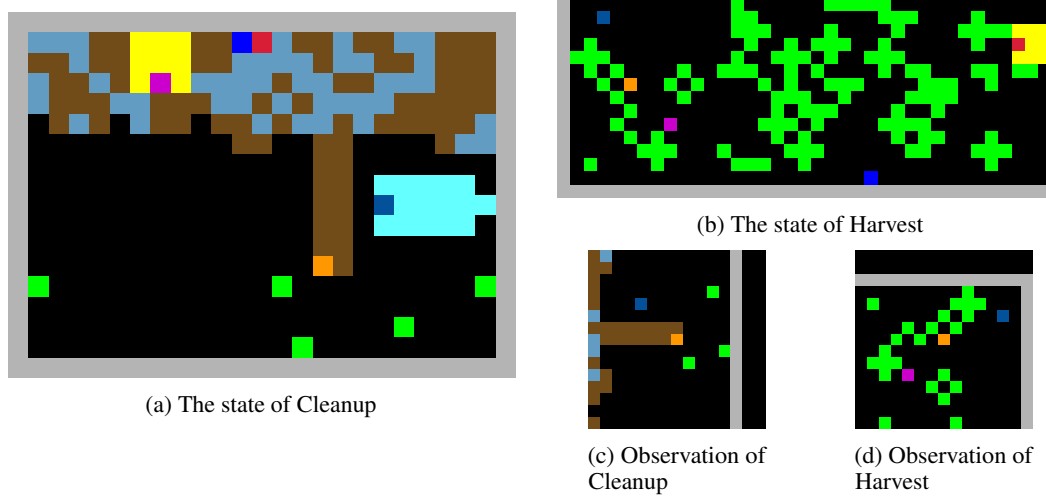

(a) The state of Cleanup

(b) The state of Harvest

(c) Observation of Cleanup

(d) Observation of Harvest

Figure 9: Global state and agent's observation of Cleanup and Harvest. In the visualization, black represents the background, green indicates apples, blue denotes the river, brown represents pollution, yellow indicates firing actions toward others, and cyan denotes cleaning actions. Other colors correspond to different agents.

Table 3: Hyperparameter settings of Cleanup and Harvest.

| Metric | Hyperparameter | Value |
|---|---|---|
| Cleanup | spawn probability of waste | 0.5 |
| | threshold for stopping the spawn of apples | 0.4 |
| | spawn probability of apple | 0.05 |
| | scarcity spawn probability of apple | 0.025 |
| Harvest | spawn probability of apple | $[0, 0.005, 0.02, 0.05]$ |
| | scarcity spawn probability of apple | $[0, 0.0025, 0.01, 0.025]$ |
| Both | episode limit | 1000 |
| | view size of agent | 7 |

growth rate of a berry type increases with its share of color in the environment, while suppressing the growth of other types. This implies that growth is slow when the three colors are balanced, whereas if all patches are red, red berries grow fastest and blue/green berries stop growing (and analogously for the other colors). Thus, beyond the planting–harvesting trade-off, agents must consider how their planting actions affect the overall environment and the competition among berry types.

In our experiments, we used 10 agents (5 preferring red and 5 preferring green to introduce an adversarial aspect). Each agent's observation covered an 11×11-pixel field of view. Each episode lasted 500 steps, and the initial state featured an equal distribution of the three berry colors across the environment.

## B  FURTHER DETAILS AND EXPERIMENTS

### B.1  FURTHER DETAILS OF THE INDEPENDENT AND DEPENDENT VALUES

The individual value $Q_t^i$ is calculated through the fusion of the independent Q-value and the dependent Q-value.

#### B.1.1  FURTHER DETAILS ON THE INDEPENDENT VALUE

The independent Q-value denotes the reward that an agent would obtain by acting independently without interacting with others, rather than being purely self-interested. Therefore, estimating the independent Q-value does not require information from other agents. Specifically, we remove

information about other agents from the global state to prevent the independent Q-value from being influenced by interactions with others.

For row-vector state inputs, we remove the dimensions corresponding to other agents, retaining only the information related to agent $i$ and the environment. For image-based matrix state inputs (Figure 9), we remove the colors corresponding to other agents and replace them with a black background. This ensures that the independent Q-value is learned without any influence from other agents.

**Rationale for Pruning and Interaction Handling.** A potential concern regarding the pruning operation is that it might introduce discontinuities in the state representation. For instance, if another agent consumes an apple within agent $i$'s field of view, the pruning operation removes the other agent, causing the apple to seemingly "vanish" from $s_t^i$, thereby affecting the estimation of $Q_{(t,\mathrm{ind})}^i$. We clarify that the disappearance of the apple is fundamentally an *interaction effect* caused by another agent. Therefore, it *should* be excluded from the independent module ($Q_{(t,\mathrm{ind})}^i$) to prevent misattribution. Instead, this interaction information is preserved in the local observation $o_t^i$ and is explicitly captured by the dependent module ($Q_{(t,\mathrm{dep})}^i$). The hypernetwork then fuses these two complementary branches, ensuring that the missing interaction signal in the independent state is compensated by the dependent component, rather than creating misleading signals.

### B.1.2 FURTHER DETAILS ON THE DEPENDENT VALUE

The dependent Q-value, in contrast to the independent Q-value, characterizes the rewards obtained through interactions with other agents, including both cooperation and defection, rather than being limited to simple "social cooperation." According to Bernstein et al. (2002); Foerster et al. (2016), agent interactions are typically restricted to neighboring agents within a certain distance. Thus, using the global state introduces unnecessary redundancy and increases computational costs, especially as the number of agents grows. To address this, we rely only on local information to improve efficiency and scalability.

We further investigated whether incorporating local historical information benefits the estimation of the dependent value. Specifically, we combined historical trajectories with the current observation when estimating the dependent Q-value and conducted additional experiments. The historical trajectories were encoded using an RNN (GRU), and we denote this variant as USE with RNN. The results are shown in Table 4. We observed that the RNN variant neither improved nor destabilized USE in Allelopathic Harvest, and slightly degraded performance in Cleanup and Harvest. This suggests that additional history is not beneficial in these settings. We attribute this to the RNN incorporating many trajectories of distant agents that are no longer within the current interaction neighborhood, thereby introducing noise when estimating the current dependent Q-value and consequently harming performance.

Table 4: Effect of Incorporating Historical Information on Dependent Value Estimation.

|  | **Cleanup** | **Harvest** | **Allelopathic Harvest** |
|---|---|---|---|
| USE | 1238.3±102.5 | 1096.0 ± 87.8 | 94.5±10.4 |
| USE with RNN | 649.1±75.3 | 326.7 ± 53.4 | 94.3±12.7 |

### B.1.3 ANALYSIS OF HYPERNETWORK INPUTS

Table 5: Performance comparison using pruned state ($s_t^i$), global state ($s_t$), and local observation ($o_t^i$) as hypernetwork inputs.

|  | $s_t^i$ | $s_t$ | $o_t^i$ |
|---|---|---|---|
| Cleanup | 18.3 ± 48.6 | 673.7 ± 91.6 | **1238.3±102.5** |
| Harvest | 186.6 ± 29.4 | 651.4 ± 73.3 | **1096.0±87.8** |

To investigate the impact of the hypernetwork's input on value fusion, we evaluated variants replacing the local observation $o_t^i$ with the pruned state $s_t^i$ and the global state $s_t$, respectively. The results in Table 5 show that $o_t^i$ yields superior performance.

The inferior performance of $s_t$ stems from the inclusion of substantial redundant global information irrelevant to the specific agent, which hinders the accurate estimation of individual value. Conversely, the poor performance of $s_t^i$ occurs because it explicitly excludes information about other agents. Lacking this interaction context, the hypernetwork fails to generate appropriate weights to effectively gate the dependent component, leading to suboptimal fusion.

## B.2 PERFORMANCE IN MORE COMPLEX ENVIRONMENTS

**Robustness.** To evaluate the robustness of USE, we conducted experiments by adding Gaussian noise to the global state in Cleanup. As shown in Table 6, USE still outperforms the other baselines, demonstrating the robustness of our approach.

**Asymmetric Adversaries.** We further evaluated our method on a variant of Allelopathic-Harvest, which involves 4 red-preferring agents and 12 green-preferring agents. The collective reward results are summarized in Table 6. In this setting, USE achieves a collective reward of 105.6, which is significantly higher than all baselines.

This indicates that even under more challenging conditions, including a larger number of agents (16), more complex game dynamics (with adversarial interactions), and more intense competition (due to scarcity), USE achieves superior performance compared to the baselines and more effectively promotes cooperation among agents.

Table 6: The results in mixed-motive environments with noise and asymmetric adversarial settings. The results show that USE surpasses all baselines and achieves better overall performance.

|  | SLI | AGA | USE |
|---|---|---|---|
| Noisy Cleanup | 244.3±29.3 | 13.4±11.9 | **550.8±79.6** |
| Asymmetric Allelopathic Harvest | 65.8±19.5 | 44.6±16.6 | **105.6±13.1** |

**Balancing Cooperation and Divergence.** A potential concern in coupling individual and collective rewards is the risk of suppressing innovative selfish behaviors that may benefit the group in the long term (i.e., favoring "safe cooperation" over "beneficial divergence"). USE addresses this trade-off through two key mechanisms. First, the VLink signal ($g_t^i$) is derived from the derivative of the *long-term* collective Q-value ($Q_t^{\text{tot}}$), rather than instantaneous rewards. Consequently, exploratory behaviors that appear selfish in the short term but eventually lead to higher collective returns are captured by the Q-value and explicitly encouraged via a positive $g_t^i$ signal. Second, VLink functions as a soft shaping term within the optimization objective (Eq. 14), which maximizes the sum of the standard individual advantage ($A_t^i$) and the weighted VLink ($\lambda g_t^i$). This structure ensures that individual incentives, essential for discovering innovative strategies, are preserved and not overridden by collective constraints. The superior performance in complex environments like Cleanup and Harvest confirms that USE successfully navigates this balance, fostering beneficial divergence while preventing destructive free-riding.

## B.3 COMPARISON OF USE WITH PURE COOPERATION AND CREDIT ASSIGNMENT

Unlike in purely cooperative tasks, agents in mixed-motive cooperation tasks typically receive individual rewards that are not fully aligned with the collective reward. Therefore, purely cooperative methods are not applicable to mixed-motive settings. In contrast to classical credit assignment, where the objective is to infer each agent's contribution to a shared global reward, mixed-motive environments already provide individually assigned rewards. However, these rewards do not accurately reflect each agent's contribution to the team. In mixed-motive settings, the reward structure is neither globally shared nor allocated based on the agents' marginal contributions. Consequently, inaccurate credit assignment may lead to the exploitation of cooperative agents, giving rise to the problem of unfairness, an issue that cannot be resolved by credit assignment alone.

To demonstrate these differences, we conducted comparisons on Cleanup and Harvest between the purely cooperative method QMIX (Rashid et al., 2020) and the credit assignment method COMA (Foerster et al., 2018). The results, summarized in Table 7, show that USE consistently outperforms both QMIX and COMA. Notably, QMIX performs poorly and even fails to achieve positive returns. We attribute this to the underlying Individual-Global-Max (IGM) assumption of

QMIX, which presumes that maximizing individual returns will lead to maximizing the global return. However, this assumption does not hold in mixed-motive settings.

Table 7: The comparison between QMIX, COMA, and USE highlights the effectiveness of USE.

|         | QMIX          | COMA        | USE             |
|---------|---------------|-------------|-----------------|
| Cleanup | -448.7±203.7  | 584.3±103.1 | **1238.3±102.5** |
| Harvest | -269.2±241.2  | 289.4±60.5  | **1096.0±87.8**  |

## B.4 THE EFFECT OF VALUE LINK

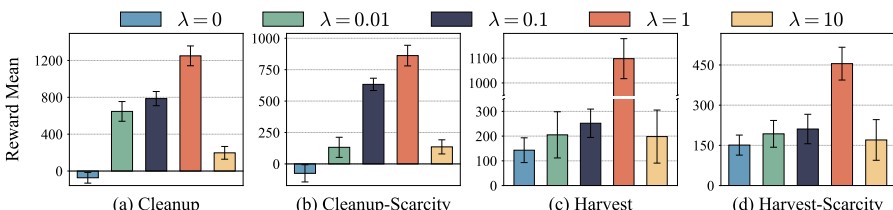

Figure 10: The impact of $g_t^i$.

To investigate the effect of Value Link ($g_t^i$), we analyze its impact on the performance. We assign different weights to $g_t^i$ in Eq. 14 by varying the value of $\lambda$, and conduct experiments accordingly. The results are presented in Figure 10. We observe that: (1) As $\lambda$ increases from 0 to 1, the performance of USE gradually improves, peaking at $\lambda = 1$. This indicates that the Value Link provides a meaningful signal regarding the contribution of agent $i$'s actions to the collective reward. We attribute the optimality of $\lambda = 1$ to the balance of magnitudes: in our settings, the norm of the VLink term ($\|g\|$) and the individual advantage ($\|A\|$) are inherently of a similar order of magnitude. Therefore, $\lambda \approx 1$ ensures a balanced optimization where neither the collective signal nor the individual advantage dominates the update, allowing $g_t^i$ to play a proper adjusting role. (2) The performance drops sharply when $\lambda = 10$. Consistent with our magnitude analysis, this occurs because an excessively large $\lambda$ causes the weighted VLink signal ($\lambda\|g\| \gg \|A\|$) to dominate the policy update, significantly weakening the influence of the individual objective. Hence, assigning a weight that aligns the scales of individual and collective signals can encourage cooperative behavior while preserving individual incentives, thereby improving overall performance.

## B.5 Q-VALUE ACCURACY AND STABILITY

To investigate the accuracy and stability of USE's Q-value estimates, we report the mean (reflecting accuracy) and standard deviation (reflecting stability) of the critic loss over five random seeds. We compare USE against MAPPO and PPO, as they utilize standard critic architectures that serve as representative baselines.

As shown in Figure 11(a), USE exhibits convergence behavior and stability comparable to MAPPO and PPO after approximately $3 \times 10^7$ steps. We note that the absolute critic loss of USE is slightly higher; however, this reflects a difference in **value scale** rather than reduced accuracy. Since USE learns a policy that achieves significantly higher returns, the corresponding target Q-values are larger in magnitude. Consequently, even with a comparable relative Temporal-Difference (TD) error, the absolute TD loss appears larger simply due to the higher numerical scale of the targets.

To validate this, Figure 11(b) presents the **relative loss**, defined as the ratio of the mean critic loss to the mean absolute Q-value: $\mathcal{L}_{\text{rel}} = \mathcal{L}_c / \mathbb{E}_{\mathcal{D}}[|Q|]$, where $\mathbb{E}_{\mathcal{D}}[|Q|]$ represents the mean absolute Q-value in the replay buffer. The results show that the relative loss for USE converges stably and is comparable to that of the baselines, confirming the accuracy and stability of its Q-value estimation despite the larger value scale.

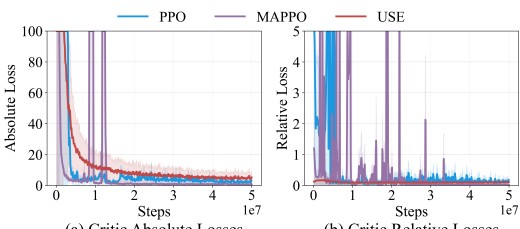

(a) Critic Absolute Losses     (b) Critic Relative Losses

Figure 11: The comparison of critic loss.

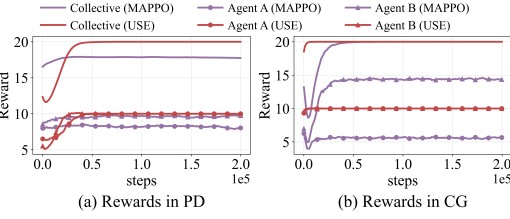

(a) Rewards in PD     (b) Rewards in CG

Figure 12: The comparison of individual rewards in PD and CG.

### B.6 FAIRNESS AND STRATEGY ANALYSIS

#### B.6.1 INDIVIDUAL REWARDS ANALYSIS: MAPPO VS. USE

To further elucidate the fairness disparity between USE and MAPPO, Figure 12 presents the training curves of individual rewards for both methods in the Prisoner's Dilemma (PD) and Chicken Game (CG).

We observe a significant reward disparity between the two agents under MAPPO, particularly in the Chicken Game (CG). This phenomenon stems from the payoff structure of CG: both mutual cooperation $(C, C)$ and unilateral defection $(C, D)$ yield an identical collective reward of 20 (payoffs: $(10, 10)$ vs. $(5, 15)$), yet their distributions differ fundamentally. Since MAPPO optimizes a purely collective objective, it is indifferent to the distribution of the total return. Consequently, it often converges to an asymmetric equilibrium where one agent is systematically exploited (receiving 5) while the other exploits (receiving 15). As this exploitative pattern still maximizes the global return, the collective learning signal provides no incentive to rectify the unfairness.

In contrast, USE explicitly incorporates the individual objective through its unified self-and-collective value design. This mechanism ensures that agents safeguard their own payoffs while promoting collective outcomes, thereby discouraging long-term systematic exploitation and achieving significantly better fairness, as evidenced by the lower Gini coefficient.

#### B.6.2 QUALITATIVE ANALYSIS OF LEARNED STRATEGIES

While quantitative results demonstrate USE's superior fairness (low Gini coefficient), understanding the underlying behavioral dynamics is crucial. In mixed-motive environments like Cleanup, a fixed division of labor (e.g., one agent permanently cleans while others harvest) would maximize collective efficiency but result in severe inequality and exploitation. USE avoids this by promoting a dynamic role-switching strategy, effectively balancing individual incentives with collective outcomes.

The mechanism driving this behavior lies in the policy gradient update rule: $\nabla J = \nabla \pi \cdot (A_t^i + \lambda g_t^i)$. Here, the update direction is determined by the interplay between the individual advantage $A_t^i$ (representing self-interest, e.g., harvesting) and the Value Link $g_t^i$ (representing the sensitivity of collective welfare to the agent's behavior). This design allows agents to promote collective cooperation while simultaneously safeguarding their individual rewards.

Specifically, if an agent continuously cleans the river, it incurs a low individual return, resulting in a low or negative individual advantage $A_t^i$. This individual signal drives the agent to switch behaviors toward harvesting to increase its own payoff. Conversely, if an agent persistently harvests while the environment degrades, the pollution accumulation reduces the long-term collective reward. This triggers a negative VLink signal ($g_t^i < 0$), penalizing the harmful harvesting behavior. The strong collective signal overrides the individual incentive, urging the agent to switch back to cleaning to restore the environment. In this manner, agents achieve a dynamic equilibrium of role switching. This mechanism maintains high collective rewards by ensuring the public good is provided, while simultaneously ensuring fairness by preventing any single agent from being permanently exploited, effectively counteracting short-term greed and preventing the tragedy of the commons.

### B.7 SENSITIVITY ANALYSIS ON PAYOFF PARAMETERS

To evaluate the robustness of USE under varying incentives, we conducted a sensitivity analysis on the Prisoner's Dilemma (PD) payoff matrix. We fixed the Reward at $R = 10$ and Punishment at $P = 5$, while systematically varying the Sucker's Payoff $S \in \{-2, 0, 2\}$ and the Temptation $T \in \{13, 15, 17\}$. This yields nine variants ranging from harsh to mild dilemmas. Figure 13 visualizes the cooperation rate for each combination.

The results demonstrate that USE achieves a consistently high cooperation rate across all variants. Specifically: (1) In most cases, the cooperation rate remains above 0.98, indicating strong robustness to parameter variations. (2) A slight decrease is observed in the most challenging setting ($T = 17, S = 2$). Here, the total payoff of unilateral defection ($T + S = 19$) is nearly identical to mutual cooperation ($2R = 20$), substantially increasing the incentive to defect. Nevertheless, the cooperation rate in this region remains high ($> 0.96$). These findings confirm that USE effectively promotes cooperation even under payoff configurations with extreme incentives for defection.

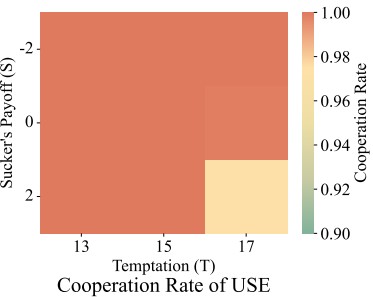

Figure 13: Cooperation rate of different parameter combinations of the Prisoner's Dilemma (PD).

Table 8: Performance of USE against an Always-Defect Opponent in the Iterated Prisoner's Dilemma.

|  | Number of cooperation | Number of defections | Cooperation rate |
|---|---|---|---|
| value | 102 | 898 | 10.2% |

### B.8 INTERACTION WITH A DEFECTING OPPONENT

To evaluate the exploitability of USE, we deployed it against an *Always Defect* opponent in the Iterated Prisoner's Dilemma for 1,000 rounds. As summarized in Table 8, USE exhibits a cooperation rate of only 10.2%. This result demonstrates that USE is not naive; it effectively identifies the consistently selfish behavior of the opponent and adapts by refraining from cooperation, thereby avoiding substantial exploitation.

### B.9 THE PERFORMANCE OF LOCAL-USE

Table 9: The comparison between MOA, IAI, LASE (Kong et al., 2024) and local-USE highlights the effectiveness of USE.

|  | MOA | IAI | LASE | local-USE |
|---|---|---|---|---|
| Cleanup | 607.3 ± 166.8 | 543.6 ± 181.3 | 617.4 ± 82.5 | **863.1 ± 62.9** |
| Harvest | 231.4 ± 60.4 | 213.0 ± 50.4 | 282.0 ± 118.9 | **360.4 ± 57.2** |

To further explore the scalability of USE under restricted information settings, we implemented a variant named **local-USE**. In this variant, the dependency on the global state $s_t$ is removed. Specifically, the input for the independent module is derived solely from the agent's local observation, where information regarding other agents is explicitly masked out. Consequently, the IDD critic estimates values based exclusively on the agent's own observation field. Table 9 reports the collective returns. We observe that local-USE maintains robust performance and significantly outperforms the baselines, demonstrating the effectiveness of our method even without global state information.

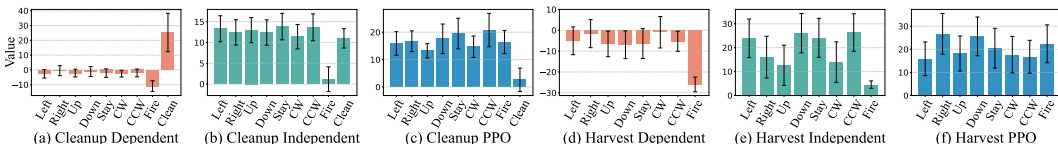

Figure 14: Visualization of converged Q-values for actions in Cleanup and Harvest.

## B.10 Q-VALUE ESTIMATION ANALYSIS

To investigate whether the Q-values learned by USE genuinely reflect the collective benefit of actions compared to baselines, we visualized the converged Q-values of USE and PPO for actions in both Cleanup and Harvest environments. The comparison is presented in Figure 14.

**Analysis in Cleanup.** We observe distinct differences in how the two methods evaluate critical actions: (1) PPO assigns a near-zero Q-value to the *Clean* action. Since cleaning yields no immediate individual reward and its long-term benefit (apple generation) is often claimed by other agents, PPO fails to credit this action, effectively discouraging cooperation. In contrast, USE's dependent module successfully captures the positive interaction value of cleaning, assigning it a substantially higher Q-value, thereby encouraging the provision of public goods. (2) PPO assigns relatively high Q-values to the *Fire* action, as defecting allows an agent to reduce competition and harvest more quickly. However, this behavior undermines collective cooperation. Conversely, USE's dependent module assigns a negative interaction value to *Fire*, reflecting its detrimental impact on the group and effectively suppressing such defection.

**Analysis in Harvest.** A similar pattern is observed in Harvest. PPO tends to overestimate the value of *Fire* due to short-term competitive gains. In contrast, USE captures the detrimental impact of this action via its dependent component by assigning it a negative value. This precise valuation effectively suppresses defective behaviors, thereby enhancing collective performance.

These empirical results confirm that USE produces Q-values that are more informative regarding collective benefits than standard baselines, validating the effectiveness of the IDD and ICC mechanisms.

## C ALGORITHMS AND HYPERPARAMETERS

### C.1 PROOF OF CONVERGENCE

**The boundedness of VLink.** $A_t^i + \lambda g_t^i$ acts as a scalar weighting factor on the policy gradient; it does not change the form of $\nabla_{\theta^i} \log \pi^i$. Provided this weight is bounded and not identically zero, each policy-gradient step remains valid and the optimization structure of $J$ is preserved. In particular, $A_t^i$ is bounded and $A_t^i + \lambda g_t^i$ is clearly not identically zero.

To show $g_t^i$ is bounded:

since each agent's reward and the independent reward are both bounded by the environment's maximum reward, i.e.,

$$|r_t| \leq R_{\max}, \quad |r_{\text{ind}}| \leq R_{\max},$$

and the discount factor $\gamma \in [0, 1)$.

So $Q_t^i$ and $Q_{\text{ind}}^i$ are both bounded:

$$|Q_t^i| \leq \frac{|R|_{\max}}{1 - \gamma}, \quad |Q_{\text{ind}}^i| \leq \frac{|R|_{\max}}{1 - \gamma}.$$

Similarly, the global Q-value is bounded:

$$|Q_t^{\text{tot}}| \leq n \cdot \frac{|R|_{\max}}{1 - \gamma}.$$

Since

$$Q_t^i = H(Q_{\text{ind}}^i, Q_{\text{dep}}^i),$$

where $H(\cdot, \cdot)$ is modeled as a linear mapping with bounded coefficients, and both $Q_t^i$ and $Q_{\text{ind}}^i$ are bounded, it follows that $Q_{\text{dep}}^i$ must also be bounded. Otherwise, $Q_t^i$ would violate its bounded range.

Given that $Q_t^{\text{tot}}$ and $Q_{\text{dep}}^i$ are both bounded and $Q_t^{\text{tot}}$ is continuously differentiable with respect to $Q_{\text{dep}}^i$, If the domain of $Q_{\text{dep}}^i$ is restricted to a compact interval (due to bounded Q-values), then by the Extreme Value Theorem (or Heine–Cantor), this continuous derivative

$$g_t^i = \frac{\partial Q_t^{\text{tot}}}{\partial Q_{(t,\text{dep})}^i}$$

attains a maximum and minimum on that interval, and is therefore bounded.

Therefore, our $g_t^i$ is guaranteed to be bounded and does not affect the theoretical optimization structure of $J$.

As shown in Section 5.4, our results in complex environments such as scarcity and crowding demonstrate that USE ultimately achieves stable convergence during training. Furthermore, as illustrated in Figure 7 of Section 5.6, the value of $g_t^i$ also stabilizes as training progresses.

**Stability under Function Approximation.** While the boundedness of $g_t^i$ is theoretically guaranteed for well-defined value functions, neural function approximation may introduce transient instabilities. To strictly ensure the *boundedness condition* required for Stochastic Approximation convergence (Kushner & Yin, 2003), we formally use a projection operator via gradient clipping and variance reduction via experience replay.

Specifically, let $\nabla_{\theta^i} J(\theta^i)$ denote the raw stochastic gradient of the objective function for agent $i$ at step $t$:

$$\nabla_{\theta^i} J(\theta^i) = \nabla_{\theta^i} \log \pi^i(a_t^i | o_t^i, \tau_t^i) \cdot \left[ A_t^i + \lambda g_t^i \right]$$

where $g_t^i = \partial Q_t^{\text{tot}} / \partial Q_{(t,\text{dep})}^i$ is the learned VLink term. To prevent unbounded updates caused by the potentially large magnitude of $g_t^i$, we apply a projection operator $\Pi_C$ that projects the gradient $\mathbf{x}$ onto an $L_2$-ball with radius $C$ (i.e., global gradient clipping):

$$\Pi_C(\mathbf{x}) = \mathbf{x} \cdot \min\left(1, \frac{C}{\|\mathbf{x}\|_2}\right)$$

The actual parameter update rule is thus defined as:

$$\theta_{t+1}^i \leftarrow \theta_t^i + \alpha \cdot \Pi_C\left(\mathbb{E}_{\mathcal{B} \sim \mathcal{D}}[\nabla_{\theta^i} J(\theta^i)]\right)$$

where $\alpha$ is the learning rate and $\mathcal{B}$ is a mini-batch sampled from the experience replay buffer $\mathcal{D}$. This formulation ensures two stability properties:

1. **Bounded Updates:** The effective update step is strictly bounded, i.e., $\|\theta_{t+1}^i - \theta_t^i\|_2 \leq \alpha C$, satisfying the boundedness assumption regardless of the instantaneous value of $g_t^i$.

2. **Variance Reduction:** The expectation $\mathbb{E}_{\mathcal{B} \sim \mathcal{D}}$ over the replay buffer mitigates the high variance of the single-sample derivative estimate $g_t^i$, aligning the update with the true gradient direction.

## C.2 Game-Theoretical Perspective on Value Decomposition

Our method shares a conceptual resonance with the theoretical framework established in CIAO (Wang et al., 2024), which applies Coalitional Affinity Games (CAG) to multi-agent learning. Specifically, CIAO provides a rigorous proof that in purely cooperative tasks (e.g., Open Ad Hoc Teamwork), decomposing the Joint Q-value ($Q_t^{\text{tot}}$) into an additive (linear) aggregation of individual and pairwise utilities is theoretically sufficient to reach the Dynamic Variational Strict Core (DVSC). This result validates the fundamental principle that value decomposition is a legitimate mechanism for ensuring stable cooperation.

However, distinct from the pure-coordination focus in CIAO, USE addresses Mixed-Motive scenarios (e.g., Social Dilemmas), where individual rationality ($Q_{(t,\text{ind})}^i$) often conflicts with collective welfare ($Q_{(t,\text{dep})}^i$). In such settings, a static linear summation is insufficient to capture the non-monotonic trade-offs required to resolve incentive conflicts (e.g., preventing exploitation or free-riding). Consequently, USE introduces a novel, non-linear framework by employing a Hypernetwork-based fusion, $Q_t^i = f_{hyper}(Q_{(t,\text{ind})}^i, Q_{(t,\text{dep})}^i)$. This structure serves as a weighted, non-linear generalization of the additive

form derived in CIAO. By dynamically adjusting the influence of the dependent component based on the state context, USE extends the applicability of value decomposition from optimizing joint team welfare in harmonic games to governing individual incentive dynamics in complex, conflicting environments.

### C.3 USE ALGORITHM

For the readers' convenience, pseudocode of USE has been summarized as Algorithm 1.

---
**Algorithm 1** Training of USE
---
1: Initialize IDD critic, actor, MLP and target networks.
2: Initialize replay buffer $\mathcal{D}$
3: **for** each episode **do**
4:     **for** each timestep $t = 0$ to $T$ **do**
5:         **for** each agent $i = 1$ to $N$ **do**
6:             Obtain the agent's observation $o_t^i$, action $a_t^i$, and global state $s_t$
7:             Remove the information of other agents to obtain the agent state $s_t^i$
8:             Calculate $Q_{(t,\text{ind})}^i$ and $Q_{(t,\text{dep})}^i$ according to Eqs. 2 and 3
9:             Calculate $Q_t^i$, $Q_t^{\text{tot}}$ according to Eqs. 5 and 6
10:           Calculate $g_t^i$ according to Eq. 7
11:         **end for**
12:         Calculate and optimize $\mathcal{L}_c$ according to Eqs. 10, 11 and 12
13:     **end for**
14:     **for** each timestep $t = 0$ to $T$ **do**
15:         **for** each agent $i = 1$ to $N$ **do**
16:             Obtain the agent's observation $o_t^i$ and action $a_{t-1}^i$
17:             Compute the actor's policy $\pi^i$ through the actor network.
18:             Calculate the advantage using Eq. 13
19:             Calculate and optimize the actor network according to Eq. 14
20:         **end for**
21:     **end for**
22:     Store the episode in replay buffer $\mathcal{D}$
23:     Sample a batch of episodes from $\mathcal{D}$
24:     Replace target parameters every $M$ episodes
25: **end for**
---

### C.4 HYPERPARAMETERS AND IMPLEMENTATION DETAILS

To provide a more detailed and effective description of our method, we include the hyperparameters of USE, which can be found in Table 10. Our model was trained on a setup with 4 NVIDIA A40 GPUs, an Intel Gold 5220 CPU, and 504GB of memory, optimized using the RMSprop optimizer (Tieleman & Hinton, 2012). Training for $10^8$ environment steps required approximately 20 hours of wall-clock time.

Table 10: Hyperparameter settings for the USE training.

| Metric | Hyperparameter | Value |
|--------|----------------|-------|
|        | $\lambda$ | 1 |
|        | $\gamma$ | 0.99 |
|        | Type of optimizer | RMSprop |
|        | Learning rate | 0.0001 |
| USE    | Batch size | 64 |
|        | Buffer size | 64 |
|        | Training steps | 100M |
|        | Target update interval | 200 |
|        | CNN kernel size | [3, 3] |
|        | MLP hidden dim | 64 |

## D  LARGE LANGUAGE MODELS USAGE

Large Language Models (LLMs) were used solely for language polishing and minor stylistic improvements. They did not contribute to the conceptual development, methodology, experiments, or analysis of this work. The authors take full responsibility for all content.

