# OpenReview forum: "USE: Enhancing Mixed-Motive Cooperation via Unified Self and Collective Rewards"
_ICLR.cc/2026/Conference — Submitted to ICLR 2026_

### Official Review · Reviewer_VQH3 · 2025-10-15

**Soundness:** 3
**Presentation:** 4
**Contribution:** 2
**Rating:** 4
**Confidence:** 5

**Summary:**

This paper introduces USE (Unifying Self and collEctive rewards), a novel method to enhance cooperation in mixed-motive multi-agent environments where balancing self-interest and collective good is crucial. Instead of treating individual and collective rewards as separate objectives, which can lead to free-riding or exploitation, USE unifies them by first decomposing an agent's individual Q-value into an "independent" component (from solitary action) and a "dependent" component (from interaction). The core of the method is the "Value Link" (VLink), which is the derivative of the collective Q-value with respect to the agent's dependent Q-value, effectively measuring if an agent's interactive behaviors are beneficial or harmful to the group. By integrating this VLink into the policy gradient update, the algorithm rewards pro-social actions and penalizes detrimental ones, thus aligning individual and collective goals to foster more effective and fair cooperation.

**Strengths:**

1. The empirical evaluation is comprehensive.  The authors structure their experiments around six distinct research questions, which helps to systematically investigate the method's performance, generalization, and fairness.  The included ablation studies offer supporting evidence for the utility of the designed components, such as the VLink mechanism and the hypernetwork.

2. The paper is well-written and generally easy to follow.  The use of the "Cleanup" game in the introduction is helpful for illustrating the core challenges of the mixed-motive setting, such as the free-rider problem.  This helps clarify the problem setting for the reader before the main technical contributions are detailed.

**Weaknesses:**

1. The paper's use of a centralized training (CTDE) paradigm is a notable limitation. It assumes access to global information that may be impractical in the realistic, decentralized scenarios that mixed-motive games aim to model. This makes the approach feel less scalable and potentially less applicable than methods designed for a fully decentralized (DTDE) setting, such as IAI, MOA and LASE [1].

2. The visualization in Figure 8 is puzzling.  The dependent (dep) module, which takes the local observation ($o_t^i$) as input, appears to only assign meaningful non-zero Q-values to the explicit interaction actions ("fire" and "clean").  However, since the agent's policy is also conditioned on $o_t^i$, the observation should contain sufficient signal to learn Q-values for other actions (e.g., movement) via the dep module as well.  The result suggests that only the independent (ind) module learns these values, which is strange. A possible, but undiscussed, explanation could lie with the hypernetwork. Since it also takes $o_t^i$ as input, is it possible that it learns to generate weights that effectively nullify or "gate" the `dep`  module's output for non-explicitly-social actions, routing the learning credit entirely to the `ind` module in those cases? And the hypernetwork architecture appears to draw inspiration from methods like QMIX (Figure 2 in USE is similar to Figure 2 in QMIX). However, in QMIX, the hypernetwork typically takes the global state $s_t$ as input to generate weights. What is the rationale for using $o_t^i$ over $s_t$ or $s_t^i$, and what would be the performance implications of changing this input?

3.  The process of "pruning" the global state $s_t$ to create the agent-specific state $s_t^i$ could introduce significant variance and misleading signals. For instance, if another agent eats an apple within the agent's field of view, the effect of pruning would be that the agent observes itself taking an action, and the apple simply vanishes from the state. It is unclear how prevalent such scenarios are and what impact this source of variance has on training stability.

4.  The paper presents quantitative results but falls short in providing a qualitative analysis of the agents' converged strategies. The fairness analysis, for example, reports a low Gini coefficient, suggesting equitable reward distribution. However, this seems to contradict the intuitive optimal strategy in Cleanup, which would be a fixed division of labor (e.g., one agent cleans, others harvest). Such a fixed-role strategy should result in a high Gini coefficient for extrinsic rewards. If, instead, agents are rotating roles, the paper does not explain how the USE mechanism overcomes the powerful learning signal that harvesting apples provides a much higher immediate reward than cleaning. Since the critic's TD-error objective would favor harvesting for both the `dep` (via observation) and `ind` (via state) inputs, a clearer explanation is needed for how the VLink signal consistently and successfully counteracts this greedy pressure to prevent a "tragedy of the commons" where all agents default to harvesting.

[1] Kong, Fanqi, et al. "Learning to balance altruism and self-interest based on empathy in mixed-motive games." Advances in Neural Information Processing Systems 37 (2024): 135819-135842.

**Questions:**

The mathematical notation for Q-value subscripts (e.g., "tot", "ind") is inconsistent across the paper (e.g., line 877 with \text{} vs. 239 without).

---

> ### Author Response · Authors · 2025-11-21
>
> **Q1:** The paper's use of a centralized training (CTDE) paradigm is a notable limitation. It assumes access to global information that may be impractical in the realistic, decentralized scenarios that mixed-motive games aim to model. This makes the approach feel less scalable and potentially less applicable than methods designed for a fully decentralized (DTDE) setting, such as IAI, MOA and LASE.
>
> **R1:** Thank you for your insightful comment. We understand that the global information you mentioned likely refers to $s_t^i$ in the IDD critic (Figure 2(a) $s_t^i$). We clarify that the $s_t^i$ is not the global state. Instead, it is a pruned state derived by removing information about other agents from the global state $\boldsymbol{s}_t$, retaining only the agent's own information and environmental information. Thus, while USE follows a CTDE framework formally, its actual information requirement is significantly lower than typical centralized methods.
>
> To further address concerns regarding scalability, we implemented a variant named local-USE, which operates without accessing the global state even during training. In this variant, the input for the IDD critic is derived from the agent's local observation, where all information regarding other agents is masked out. The table below reports the collective returns (mean ± std). We observe that local-USE maintains robust performance and significantly outperforms the baselines.
>
>
> |  | local-USE  |  LASE  |  MOA  |  IAI  |
> | --- | :-----:  | :----:  | :----:  | :----:  |
> | Cleanup       | **863.1±62.9** | 617.4±82.5 | 607.3±166.8 | 543.6±181.3 |
> | Harvest        | **360.4±57.2** | 282.0±118.9 | 231.4 ± 60.4 | 213.0 ± 50.4 |
>
> Regarding the mentioned baselines (IAI, MOA, LASE), they are also not strictly "DTDE" in the sense of only using self-observation and self-reward, but instead rely, to varying degrees, on cross-agent global information:
> - IAI utilizes the private rewards of other agents: *In IAI, each agent's subjective reward is defined over the smoothed rewards of all players, and the authors explicitly state in Section 3.2: "furthermore, we allow agents to observe the smoothed reward of every player on each timestep." This assumes that agents can continuously observe other agents' rewards.*
> - MOA utilizes the joint actions of all agents: *In MOA, the authors describe it as follows: "The MOA … is trained to predict all other agents' next actions given their previous actions, and the agent's egocentric view of the state: $p(\boldsymbol{a}_{t+1}|\boldsymbol{a}_t,s_t^k)$." Thus, while the optimization is decentralized (no centralized critic), the information structure still relies on a joint-action broadcast.*
> - LASE utilizes the joint actions of all agents: *In LASE, the Social Relationship (SRI) value function $Q^i_{\text{SR}}(o_i,\boldsymbol{a})$ takes the joint action $\boldsymbol{a}_t = (a_t^1,\dots,a_t^N)$ as input. In Section 4.2, Social Relationships Inference (SRI), Equation (3) explicitly uses this global joint action to compute the agents' gifting weights. LASE also uses these joint actions for counterfactual estimation.*
>
> In realistic mixed-motive scenarios, environmental information is often more accessible than the private actions or rewards of others. For instance, in pollution control, downstream agents cannot easily observe whether an upstream factory is "secretly discharging" (Hidden Action), but they can readily monitor water quality (Observable Environment). Consequently, USE demonstrates superior applicability and scalability in such realistic settings compared to methods relying on explicit action or reward sharing.
>
>
> **Q2:** The visualization in Figure 8 shows that the dependent module assigns near-zero values to independent actions (e.g., movement) despite the local observation containing sufficient signal, which suggests a potential failure to capture these values. Is it possible that the hypernetwork learns to "gate" the module's output for non-social actions?
>
> **R2:** Thank you so much for the insightful feedback! We wholeheartedly agree with your observation that the hypernetwork likely adjusts its generated weights during the optimization process. This mechanism allows the model to effectively guide the learning of non-explicit interaction actions toward the desired semantics, while simultaneously "suppressing" the learning of these actions in contexts where they are not relevant. We view the hypernetwork as a powerful tool that facilitates a seamless and dynamic division of responsibilities across the modules, ensuring that the dependent module focuses on explicit interaction actions, while the independent module specializes in self-related actions.

---

> > ### Author Response · Authors · 2025-11-21
> >
> > **Q3:** The hypernetwork architecture appears to draw inspiration from methods like QMIX (Figure 2 in USE is similar to Figure 2 in QMIX). However, in QMIX, the hypernetwork typically takes the global state dep, ind $\boldsymbol{s}_t$ as input to generate weights. What is the rationale for using $o_t^i$ over $\boldsymbol{s}_t$ or $s_t^i$, and what would be the performance implications of changing this input?
> >
> > **R3:** We appreciate the reviewer's question. First, we would like to clarify that although the architecture illustrated in Figure 2 of USE appears superficially similar to the hypernetwork mixer in QMIX, their design goals differ fundamentally. QMIX uses a hypernetwork that takes the global state $\boldsymbol{s}_t$ as input in order to enforce the IGM (Individual–Global–Max) monotonicity constraint to estimate the global action value $Q_t^{\text{tot}}$​. In contrast, USE employs a hypernetwork to dynamically fuse the independent and dependent Q-values of a single agent to estimate its individual action value.
> >
> > Regarding the choice of input, we deliberately use the local observation $o_t^i$ rather than the global state $\boldsymbol{s}_t$​. This is because using the global state introduces unnecessary redundancy and increases computational costs, especially as the number of agents grows. Using $o_t^i$​ makes the hypernetwork input dimension independent of $N$, thereby improving scalability.
> >
> > And if we were to feed $s_t^i$​ (i.e., the pruned state where all other agents' information has been removed) into the hypernetwork instead, it would not contain any signal about whether agent $i$ is currently in an interaction context (e.g., another agent is nearby so that firing or cleaning meaningfully affects others). In that case, the hypernetwork would have no basis to learn when to increase or decrease the weight assigned to the dependent component $Q^{i}_{(t,\text{dep})}$​. In other words, $s_t^i$ cannot provide the necessary gating information for interaction-dependent behavior.
> >
> > To further address your concern, we conducted additional experiments in which we replaced the hypernetwork input $o_t^i$​ with $s_t^i$ and $\boldsymbol{s}_t$, respectively; the results are reported in the following Table. We observe that using the local observation $o_t^i$ as input yields the best performance, the global state $\boldsymbol{s}_t$ is second, and the pruned state $s_t^i$​ performs worst. A possible reason is that $\boldsymbol{s}_t$​ contains substantial redundant global information that is not directly useful for agent i, which can hinder accurate estimation of its individual value, whereas $s_t^i$​ removes precisely the crucial information about other agents and thus fails to effectively gate and fuse the independent and interaction-dependent components. The related discussion has been added to Appendix B.1.3 in the revised version of the paper.
> >
> > |  | $s_t^i$  |  $\boldsymbol{s}_t$  |  $o_t^i$  |
> > | --- | :-----:  | :----:  | :----:  |
> > | Cleanup       | 18.3±48.6 | 673.7±91.6 | **1238.3 ± 102.5** |
> > | Harvest        | 186.6±29.4  | 651.4±73.3 | **1096.0 ± 87.8** |
> >
> > **Q4:** The process of "pruning" the global state $\boldsymbol{s}_t$ to create the agent-specific state $s_t^i$ could introduce significant variance and misleading signals. For instance, if another agent eats an apple within the agent's field of view, the effect of pruning would be that the agent observes itself taking an action, and the apple simply vanishes from the state. It is unclear how prevalent such scenarios are and what impact this source of variance has on training stability.
> >
> > **R4:** We would first like to clarify that the pruning operation $s_t^i = \text{Prune}(\boldsymbol{s}_t)$ is only used in the independent module, in order to avoid mistakenly attributing interaction effects to "independent" returns. The dependent module instead relies on the local observation $o_t^i$ to estimate the dependent value. In your example where another agent eats an apple within the field of view and the apple "disappears", this event is indeed an interaction effect. It is intentionally excluded from the independent module by pruning, and its influence is captured in the dependent module via $o_t^i$. The hypernetwork then fuses these two branches into the final individual value, so the missing interaction signal in $s_t^i$​ is compensated by the dependent component rather than systematically misguiding the independent component.
> >
> > We have incorporated this clarification into Appendix B.1.1.

---

> > > ### Author Response · Authors · 2025-11-21
> > >
> > > **Q5:** The paper lacks qualitative analysis of converged strategies. The reported low Gini coefficient contradicts the intuitive optimal strategy in Cleanup (fixed division of labor), which would imply high inequality. If agents rotate roles, how does USE overcome the strong immediate reward bias for harvesting favored by the critic? Please clarify how the VLink signal counteracts this greedy pressure to prevent a "tragedy of the commons" where all agents default to harvesting.
> > >
> > > **R5:** First, we would like to clarify that this paper focuses on the mixed-motive cooperation setting, where agents must balance individual incentives with collective outcomes. Under this setting, our goal is to encourage agents to cooperate in achieving higher collective rewards while still ensuring that each agent can secure its own reasonable share of the payoff.
> > >
> > > As you pointed out, a fixed division of labor (e.g., one agent always cleaning while others always harvest) would indeed lead to a high Gini coefficient and correspond to an exploitative pattern, which is precisely what we aim to avoid in mixed-motive cooperation settings (maintaining high fairness while promoting cooperation to sustain high collective rewards).
> > >
> > > Our USE achieves the goal by following design: The policy is updated via the gradient $\nabla J = \nabla \pi \cdot (A_t^i + \lambda g_t^i)$ (Equation 14). Here, $g_t^i$ (VLink) encourages agents to engage in collective cooperation (e.g., cleaning), while $A_t^i$ represents the advantage of individual reward (e.g., harvesting). This allows us to promote collective cooperation while simultaneously safeguarding individual rewards in mixed-motive environments.
> > >
> > > Specifically, if an agent continuously cleans the river, it incurs low individual returns (i.e., low $A_t^i$), which drives the agent to switch to harvesting to increase its individual reward. Conversely, if an agent persistently harvests, causing pollution accumulation and reducing the collective reward, this triggers a negative VLink signal (penalizing the harmful behavior), thereby urging the agent to switch to cleaning. In this manner, agents achieve dynamic role switching to maintain high collective rewards while ensuring fairness, effectively counteracting the pressure of short-term greed and preventing the tragedy of the commons.
> > >
> > > We have incorporated this clarification into Appendix B.6.2.
> > >
> > > **Q6:** The mathematical notation for Q-value subscripts (e.g., "tot", "ind") is inconsistent across the paper (e.g., line 877 with \text{} vs. 239 without).
> > >
> > > **R6:** Thank you for your careful review. We have made corrections in the corresponding parts of the paper and will uniformly use the \text{} format.

---

> > > > ### Author Response · Authors · 2025-11-25
> > > > **Follow-up: Do Our Responses Resolve Your Main Concerns?**
> > > >
> > > > Dear Reviewer VQH3,
> > > >
> > > > We appreciate your time and effort in reviewing our paper. We have provided detailed responses to your insightful comments and concerns, which we hope have sufficiently addressed the points raised. Some of the key clarifications include the following:
> > > >
> > > > 1. **Scalability:** We explained how USE, despite following a CTDE framework, reduces its reliance on global information, and introduced the local-USE variant that operates entirely on local observations to improve scalability. Our experiments show that local-USE performs robustly and outperforms the baselines, as shown in the provided comparison table.
> > > >
> > > > 2. **Hypernetwork Mechanism:** We clarified that the hypernetwork in USE learns a gating mechanism over the dependent module, suppressing the dependent component for non-social actions, encouraging independent modules to learn non-social actions, and that using local observations as input (instead of global states as in QMIX) leads to better performance.
> > > >
> > > > 3. **Role Switching and Fairness:** We elaborated on how the VLink signal encourages agents to dynamically switch roles, preventing the "tragedy of the commons" and maintaining fairness while promoting cooperation.
> > > >
> > > > We sincerely hope that these clarifications have resolved your concerns. If they have, we kindly request that you consider adjusting your evaluation score to reflect these updates. We highly value your feedback, as it continues to help us improve the quality of our work.
> > > >
> > > > Should you have any further questions or comments, please feel free to share them with us. We look forward to your further feedback.
> > > >
> > > > Best regards,
> > > >
> > > > The Authors

---

### Official Review · Reviewer_h3Yf · 2025-10-23

**Soundness:** 3
**Presentation:** 3
**Contribution:** 3
**Rating:** 6
**Confidence:** 4

**Summary:**

This paper introduces USE (Unifying Self and collEctive rewards), a novel framework for improving mixed-motive cooperation in multi-agent reinforcement learning (MARL), where agents must balance self-interest and collective welfare. Unlike previous approaches that treat individual and collective rewards as independent and simply optimize their weighted sum, USE decomposes each agent’s individual Q-value into an independent component (self-driven) and a dependent component (arising from interactions with others). It then correlates the dependent component with the collective Q-value through a differentiable measure called the Value Link (VLink), aligning both objectives at the value-function level. This coupling ensures that improving individual returns naturally promotes collective outcomes, mitigating problems like free-riding and exploitation. Experiments across standard social-dilemma benchmarks (Prisoner’s Dilemma, Chicken Game, Cleanup, Harvest, and Allelopathic Harvest) show that USE consistently outperforms prior methods in both performance and fairness. The study also demonstrates that the VLink can serve as a measurable indicator of agents’ cooperative tendencies.

**Strengths:**

1. The paper targets a long-standing weakness in mixed-motive MARL — the naive weighted-sum treatment of individual vs. collective rewards — and replaces it with a structured correlation mechanism. This represents a meaningful theoretical step beyond existing “reward reweighting” or “social influence” methods, which still assume separable objectives.
2. The Individual Dependence Decomposition (IDD) neatly separates the self-driven and interaction-driven components of each agent’s Q-value, providing an interpretable and modular formulation. The Individual Collective Coupling (ICC) with the derivative-based Value Link (VLink) offers a mathematically clean and differentiable way to quantify how an agent’s interaction behaviour affects group value — a novel analytic handle on cooperation.
3. By embedding the cooperative coupling term $g_i = \partial Q_{tot}/\partial Q_i^{dep}$​ directly into the policy gradient, the method avoids ad-hoc reward shaping and maintains theoretical coherence with standard MARL optimization. The inclusion of $A_i + \lambda g_i$​ preserves convergence guarantees, a rigor often missing in “social incentive” literature.
4. USE consistently outperforms six strong baselines (PPO, MAPPO, IAI, MOA, SLI, AGA) across diverse environments — from simple matrix games to long-horizon social dilemmas and adversarial mixed-motive tasks (Allelopathic Harvest). Results are robust to noise, scarcity, and population scaling (10 agents), suggesting generalization rather than over-fitting to specific dynamics.
5. The VLink offers a quantifiable proxy for cooperative tendency, observable through its correlation with prosocial actions (“clean”) and inverse correlation with antisocial actions (“fire”). The decomposition yields visualizable Q-value maps distinguishing dependent vs. independent contributions — rare interpretability in multi-agent RL.
6. The paper maintains consistency between intuition, architecture, and gradient formulation, supported by convergence analysis and boundedness proofs. The overall narrative — from social-dilemma motivation to algorithmic realization — is coherent and well-structured.
7. The authors systematically remove or alter key components (−VLink, −Hyper, −TLoss, etc.) and show predictable, interpretable degradation — reinforcing that each design choice contributes functionally rather than heuristically.

**Weaknesses:**

1. The method hinges on decomposing the individual reward into two components—“independent” vs. “interaction-dependent”. This decomposition requires the ability to prune other agents’ influence (for the independent part) and accurately capture the local interaction effects (for the dependent part). In many real-world or large-scale multi-agent systems such clean separation may not hold or may be very noisy.
2. The architecture uses a hypernetwork to fuse independent and dependent Q-values, and the Value Link (VLink) uses a partial derivative $\partial Q_{tot} / \partial Q_i^{dep}$​ to compute an agent’s contribution. These components assume differentiability, smoothness and manageable dimensionality. In very large, heterogeneous, non-stationary agent populations, or when observation spaces are huge, these assumptions may break down. Moreover, the experiments—while reasonably broad—still reside in more controlled simulated environments (e.g., “Cleanup”, “Harvest”, “Allelopathic Harvest”). Real world mixed-motive settings (with asynchronous agents, partial observability, communication issues) may behave quite differently.
3. By strongly coupling individual and collective rewards via the dependent component, there is a risk of suppressing innovative selfish behaviour that could benefit the group in the long term but doesn’t immediately align with collective Q-value. In other words, the approach may favour “safe cooperation” over “risky but beneficial divergence”. The paper does not deeply probe this trade-off in environments where individual novelty leads to collectively beneficial emergence.

**Questions:**

> 1. Please address the weaknesses.

> 2. The idea of structuring an individual agent's Q-value into two parts, i.e., $Q_i = f_{hyper}(Q_i^{ind}, Q_i^{dep})$, is novel in the literature of mixed-motive MARL. However, this idea has emerged in [1] that derives an additive form to aggregate $Q_i^{ind}$ and $Q_i^{dep})$, underpinned by Coalitional Affinity Game and prove the validity of this shaping, with connection to stable cooperation. The proposed $Q_i = f_{hyper}(Q_i^{ind}, Q_i^{dep})$ is a weighted form, generalizing the result from [1]. For this reason, I believe it should be with a discussion in this paper between the proposed structure of $Q_i = f_{hyper}(Q_i^{ind}, Q_i^{dep})$ and the linear shaping in [1]. Further, this could also underlie this paper with a game-theoretical foundation.

[1] Wang, Jianhong, Yang Li, Yuan Zhang, Wei Pan, and Samuel Kaski. "Open Ad Hoc Teamwork with Cooperative Game Theory." In International Conference on Machine Learning, pp. 50902-50930. PMLR, 2024.


> 3. Please address the following technical questions:

(1) The update rule (Eq. 14) uses $\nabla_{\theta_i} J = \nabla_{\theta_i} \log \pi_i(A_i + \lambda g_i)$.  How sensitive is training to the hyperparameter $\lambda$?  Does a fixed λ = 1 generalize across tasks, or does it require tuning?  The ablation (Appendix B.4) shows large performance drops for λ > 1 or = 0 — is there any theoretical guidance for its choice?

(2) Since $Q_{\text{tot}}$​ is computed from all agents’ $Q_i$​ (Eq. 6), doesn’t calculating $g_i^t = \partial Q_{\text{tot}}/\partial Q_i^{(\text{dep})}$​ back-propagate gradients through other agents’ critics, implicitly sharing parameters across agents during centralized training?  How does USE prevent information leakage or gradient interference between agents in decentralized execution?

(3) The convergence “proof” (Appendix C.1) relies on boundedness and non-zero weighting of $A_i+\lambda g_i$​.  But $g_i$ is itself learned and may be unbounded (e.g. with an etremely large value in practice) under certain activation functions.  Can the authors formally guarantee stability (e.g., in the sense of stochastic approximation theory) under function-approximation noise?

(4) Figure 7 claims that $g_i^t$​ correlates with “clean” vs. “fire” actions, but can $g_i^t$ ever flip sign within a single trajectory due to noise or network instability?  If yes, how does the algorithm prevent mis-classification of cooperative actions during training?

---

> ### Author Response · Authors · 2025-11-21
>
> **Q1:** Components—"independent" vs. "interaction-dependent". This decomposition requires the ability to prune other agents' influence (for the independent part) and accurately capture the local interaction effects (for the dependent part). In many real-world or large-scale multi-agent systems such clean separation may not hold or may be very noisy.
>
> **R1:** We fully agree with the reviewer that in complex or large-scale systems, achieving a perfectly clean separation between "independent" and "interaction-dependent" signals is inherently difficult and often noisy. However, we clarify that a perfect physical isolation is not a strict prerequisite for USE.
>
> To explicitly address your concern regarding noisy separation, we conducted additional experiments under noisy-state conditions (detailed in Appendix B.2 and Table 6), where input states were corrupted to deliberately hinder clean decomposition. The results below indicate that USE remains robust and continues to outperform baselines even when the separation boundary is blurred. This confirms that our framework is tolerant to decomposition errors and does not require a pristine, noise-free separation to improve cooperation.
>
> |  | SLI  |  AGA  |  USE  |
> | --- | :-----:  | :----:  | :----:  |
> | Noisy Cleanup | 244.3±29.3 | 13.4±11.9 | **550.8±79.6** |
>
>
> **Q2:** The architecture relies on hypernetworks and partial derivatives, assuming differentiability and manageable dimensionality. How does it handle challenges such as very large, heterogeneous, or non-stationary populations, and huge observation spaces?
>
> **R2:** Thank you for your comment. In the following, we will discuss how USE handle these changlles:
> 1. **Huge Observation Spaces:** As shown in Figure 2, we employ standard deep neural networks (CNNs/MLPs) in USE (IDD critic, hypternetwork and actor) to encode high-dimensional observations into compact embeddings. This effectively addresses the issue of huge observation spaces.
> 2. **Large Populations:** As shown in Figure 2, the the critic of  USE operate solely on local observations ($o_t^i$), pruned states ($s_t^i$) and acions ($a_t^i$), while the actor of USE operate only on local observations ($o_t^i$) and the action from the previous timestep ($a_{t-1}^i$). This means that the input to USE is independent of the population size $N$, which helps maintain computational stability at scale.
> 3. **Heterogeneous Populations:** As shown in Figure 2, the hypernetwork generates dynamic fusion parameters ($W, b$) conditioned on agent-specific observations. This enables the model to learn adaptive behaviors for heterogeneous agents and agent-specific strategies.
> 4. **Non-stationary Populations:** We mitigate instability by treating VLink ($g_t^i$) as a bounded (Appendix C.1), detached scalar coefficient. Combined with target networks and gradient clipping, this effectively dampens noise and variance arising from the volatile dynamics of other agents.
>
> **Q3:** The experiments—while reasonably broad—still reside in more controlled simulated environments (e.g., "Cleanup", "Harvest", "Allelopathic Harvest"). Real world mixed-motive settings (with asynchronous agents, partial observability, communication issues) may behave quite differently.
>
> **R3:** Thank you for your questions. First of all, Our USE uses a centralized training decentralized execution (CTDE) framework, which means that our agents act independently once trained. Agents relies only on the local observations during execution, without the need for communication, which allows USE to naturally adapt to issues such as partial observability and asynchronous communication in real-world environments.
>
> To further approximate real-world complexities, we explicitly stress-tested USE under harsh conditions, such as introducing Gaussian noise to simulate sensor errors (detailed in Table 6) , reduced resources (Scarcity, Figure 5) and increased agent density (Crowding, Figure 5). The consistent performance gains observed across these diverse settings demonstrate that USE is robust to environmental variations.

---

> > ### Author Response · Authors · 2025-11-21
> >
> > **Q4:** By strongly coupling individual and collective rewards via the dependent component, there is a risk of suppressing innovative selfish behaviour that could benefit the group in the long term but doesn't immediately align with collective Q-value. In other words, the approach may favour "safe cooperation" over "risky but beneficial divergence". The paper does not deeply probe this trade-off in environments where individual novelty leads to collectively beneficial emergence.
> >
> > **R4:** Thank you for raising this point. We hypothesize that the collective Q-value you refer to may correspond to the collective reward. Generally speaking, an action that benefits the long-term collective tends to increase joint Q-value ($Q_t^{\text{tot}}$).
> >
> > We would like to clarify that USE encourages what might be perceived as 'innovative selfish behavior' due to its optimization focus on long-term value (Q-value) rather than immediate rewards, with the coupling mechanism (VLink) serving as a soft shaping term. In the following, we will provide a detailed explanation.
> >
> > The policy is updated via the gradient $\nabla_{\theta^{i}}J(\theta^{i}) = \nabla_{\theta^{i}}\log \pi^{i}(a_{t}^{i}|o_{t}^{i},\tau_{t}^{i}) \cdot [A_{t}^{i} + \lambda g_{t}^{i}]$, where the goal is to maximize the combined term $[A_{t}^{i} + \lambda g_{t}^{i}]$. Here, $A_{t}^{i}$ represents the standard individual advantage derived from the individual long-term Q-value, and $g_{t}^{i}$ (Value Link) is defined as the partial derivative of the long-horizon collective Q-value $Q_{t}^\text{tot}$ with respect to the dependent component, i.e., $g_{t}^{i} = \partial Q_{t}^\text{tot} / \partial Q_{(t,\text{dep})}^{i}$. Crucially, both terms are computed based on cumulative discounted returns, capturing the long-term consequences of actions. If an action could benefit the group in the long term but doesn't immediately align with collective rewards, the corresponding $A_{t}^{i}$ and $g_{t}^{i}$ would contribute to maximizing the agent's objective. Therefore, USE agents are inclined to take this action.
> >
> > We have included the discussion on this part in Appendix B.2.
> >
> > **Q5:** The idea of structuring an individual agent's Q-value into two parts, i.e., $Q_t^i=f_{hyper}(Q^i_{(t, \text{ind})},Q^i_{(t, \text{dep})})$, is novel in the literature of mixed-motive MARL. However, this idea has emerged in [1] that derives an additive form to aggregate $Q^i_{(t, \text{ind})}$ and $Q^i_{(t, \text{dep})}$, underpinned by Coalitional Affinity Game and prove the validity of this shaping, with connection to stable cooperation. The proposed $Q_t^i=f_{hyper}(Q^i_{(t, \text{ind})},Q^i_{(t, \text{dep})})$ is a weighted form, generalizing the result from [1]. For this reason, I believe it should be with a discussion in this paper between the proposed structure of $Q_t^i=f_{hyper}(Q^i_{(t, \text{ind})},Q^i_{(t, \text{dep})})$ and the linear shaping in [1]. Further, this could also underlie this paper with a game-theoretical foundation.
> >
> > [1] Wang, Jianhong, Yang Li, Yuan Zhang, Wei Pan, and Samuel Kaski. "Open Ad Hoc Teamwork with Cooperative Game Theory." In International Conference on Machine Learning, pp. 50902-50930. PMLR, 2024.
> >
> > **R5:** Thank you for your suggestion. We acknowledge that [1] (CIAO) provides rigorous proof that decomposing the Joint Q-value into additive components is theoretically valid for ensuring stability in Coalitional Affinity Games (CAG). We have revised the paper to discuss the connection and differences with [1] (CIAO).
> > Notably, there are two fundamental differences between USE and CIAO:
> > 1. **Problem Scope:** CIAO focuses on **Pure Cooperation**, where linear decomposition suffices for team stability. In contrast, USE addresses **Mixed-Motive Cooperation**, where individual and collective interests often conflict. We employ a hypernetwork-based non-linear fusion ($Q_t^i = f_{hyper}(\dots)$) as an architectural innovation to capture the non-monotonic trade-offs (e.g., reducing cooperation if exploited) required to resolve these conflicts, which static linear forms cannot represent.
> > 2. **Decomposition Target:** CIAO performs **Global Q-Value Decomposition** ($Q^{\text{tot}}$) to coordinate the team. Conversely, USE performs **Individual Q-Value Decomposition** ($Q_t^i$) into independent and dependent components. This distinction is vital in mixed-motive settings where agents do not optimize a single global reward; instead, each agent must strike a balance between individual rewards and collective rewards.
> >
> > By framing the linear form in CIAO as a special case for pure cooperation, we position USE as a parallel advancement that adapts value decomposition to the complex dynamics of mixed-motive games.

---

> > > ### Author Response · Authors · 2025-11-21
> > >
> > > **Q6:** The update rule ( Equation. 14) uses $\nabla_{\theta^i} J(\theta^i) =  \nabla_{\theta^i} \log \pi^i(a_t^i|o_{t}^i,\tau_t^i) \cdot \left[A^i_t + \lambda g_t^i \right]$. How sensitive is training to the hyperparameter $\lambda$? Does a fixed λ = 1 generalize across tasks, or does it require tuning? The ablation (Appendix B.4) shows large performance drops for $\lambda$ > 1 or = 0 — is there any theoretical guidance for its choice?
> > >
> > > **R6:** We thank the reviewer for this question. Empirically, we found that a fixed $\lambda = 1$ consistently yields the best results across all evaluated environments, indicating strong generalization without the need for task-specific tuning. A possible theoretical explanation for this choice lies in the balance of magnitudes: in our settings, the norms of the VLink term ($\|g\|$) and the individual advantage ($\|A\|$) are inherently of a similar order of magnitude. Consequently, setting $\lambda \approx 1$ ensures a balanced optimization where neither the collective signal nor the individual advantage dominates the update, allowing $g_t^i$ to play a proper adjusting role. In contrast, extreme values such as $\lambda=10$ or $\lambda=0$ cause one of the two terms to excessively dominate the gradient direction, leading to the performance degradation observed in the ablation study. We have included this analysis of magnitude balancing in Appendix B.4.
> > >
> > > **Q7:** Since $Q_t^\text{tot}$​ is computed from all agents' $Q_t^i$​ (Eq. 6), doesn't calculating $g_t^i=\partial Q_t^\text{tot} / \partial Q_{(t,\text{dep})}^i$​ back-propagate gradients through other agents' critics, implicitly sharing parameters across agents during centralized training (1)? How does USE prevent information leakage or gradient interference between agents in decentralized execution (2)?
> > >
> > > **R7:** As described in Equation 7, $g_t^i$ is computed as the partial derivative of the $Q_t^\text{tot}$ with respect to the dependent value $Q_{(t, \text{dep})}^i$.
> > >
> > > **(1)** We clarify that during this computation, we treat the Q-values of all other agents as constants (i.e., we detach their gradient flows). This explicitly prevents the computation of $g_t^i$ from causing gradients to backpropagate through the critic networks of other agents.
> > > $$
> > > g_t^i = \lim_{\Delta Q^i_{(t, \text{dep})} \to 0}
> > > \frac{\Delta Q_t^{\text{tot}}}{\Delta Q_{(t, \text{dep})}^i}
> > > = \frac{\partial Q_t^{\text{tot}}}{\partial Q_{(t, \text{dep})}^i}
> > > $$
> > > Furthermore, $g_t^i$ is utilized solely as a scalar coefficient for the agent's optimization objective and does not participate in the critic's own backpropagation process. This design guarantees that the use of $g_t^i$ does not introduce any implicit parameter sharing during training phase.
> > >
> > > **(2)** As presented in Equation 14,
> > > $$\nabla_{\theta^i} J(\theta^i) =  \nabla_{\theta^i} \log \pi^i(a_t^i|o_{t}^i,\tau_t^i) \cdot \left[A^i_t + \lambda g_t^i \right],$$
> > >
> > > regarding the agent's policy update, $g_t^i$ serves as a stop-gradient scalar coefficient, implying that it does not propagate gradients back to the agent's network. Consequently, there is no risk of information leakage or gradient interference during the fully decentralized execution phase.

---

> > > > ### Author Response · Authors · 2025-11-21
> > > >
> > > > **Q8:** The convergence "proof" (Appendix C.1) relies on boundedness and non-zero weighting of $A_i+\lambda g_t^i$​. But $g_t^i$ is itself learned and may be unbounded (e.g. with an extremely large value in practice) under certain activation functions. Can the authors formally guarantee stability (e.g., in the sense of stochastic approximation theory) under function-approximation noise?
> > > >
> > > > **R8:** Thank the reviewer for your insightful comment. To provide a formal guarantee, we have revised Appendix C.1 to explicitly incorporate the stability mechanisms inherent in our implementation.
> > > >
> > > > Formally, we model the Global Gradient Clipping employed in our algorithm as a Projection Operator $\Pi_C$, which projects the computed gradient onto an $L_2$-ball of radius $C$. Mathematically, this operator ensures that the effective parameter update step is strictly bounded, satisfying $\|\theta_{t+1}^i - \theta_t^i\|_2 \le \alpha C$. By enforcing this constraint, the algorithm satisfies the boundedness condition required for the almost sure convergence of Robbins-Monro type algorithms, thereby preventing divergence regardless of the instantaneous magnitude of the learned term $g_t^i$.
> > > >
> > > > Furthermore, to mitigate the function approximation noise highlighted by the reviewer, the algorithm utilizes an Experience Replay Buffer. As formalized in the revised proof, the expectation over the mini-batch $\mathbb{E}_{\mathcal{B} \sim \mathcal{D}}$ serves to approximate the true gradient direction and reduce the variance of the derivative estimate $g_t^i$, aligning the update with the i.i.d. sampling assumptions of Stochastic Approximation (SA) theory.
> > > >
> > > > Empirically, this formal analysis is supported by Figure 7, which demonstrates that $g_t^i$ stabilizes within a bounded range after the initial warm-up phase, and Figure 10, which confirms that the algorithm maintains stability even under a large weighting coefficient ($\lambda=10$).
> > > >
> > > > **Q9:** Figure 7 claims that $g_t^i$​ correlates with "clean" vs. "fire" actions, but can $g_t^i$ ever flip sign within a single trajectory due to noise or network instability? If yes, how does the algorithm prevent mis-classification of cooperative actions during training?
> > > >
> > > > **R9:** Thank you for this insightful question. Yes, your statement is correct.. The $g_t^i$ is defined as $(g_t^i = \partial Q_t^{\text{tot}} / \partial Q^i_{(t, \text{dep})}) (Eq. (7))$, and as such, its value is closely related to the accuracy and stability of the Q-values, which are directly determined by the critics. In the early stage of training, when the critics for $Q^i_{(t, \text{dep})}$ and $Q_t^{\text{tot}}$ are still noisy, the $g_t^i$ can indeed fluctuate and occasionally change sign even within a single trajectory. However, as the training progresses and the critics converge, $g_t^i$ also becomes more accurate and stable. Therefore, it does not systematically misclassify cooperative actions as defective ones and does not affecting the final performance.

---

> > > > > ### Comment · Reviewer_h3Yf · 2025-11-21
> > > > >
> > > > > I appreciate the authors' detailed and clear responses to my questions. My concerns have been removed and I now confirm that this is a paper worth being published. I decide to raise my score to 8.

---

### Official Review · Reviewer_8Ba6 · 2025-10-28

**Soundness:** 2
**Presentation:** 3
**Contribution:** 3
**Rating:** 4
**Confidence:** 3

**Summary:**

The core objective of this paper is to address the issues of inaccurate credit assignment and objective conflict in multi-agent mixed-motive cooperation tasks. Conventional methods typically optimize by maximizing the simple weighted sum of individual and collective rewards, $R = \lambda R_{individual} + (1-\lambda) R_{collective}$. The authors point out that this design can lead agents to converge to suboptimal policies, skewed towards either extreme selfishness or extreme altruism, failing to achieve a true dynamic balance between the two objectives. To overcome this limitation, the paper proposes the Unified Self and Collective Rewards (USE) method. USE aims to deeply fuse individual and collective interests, rather than simply superimposing them, through a novel reward structure or policy objective. This enables agents to more effectively learn an equilibrium strategy that both maximizes their own returns and promotes overall collaboration. The method emphasizes establishing an intrinsic link between individual and collective behavior in the reward signal design, guiding agent actions and enhancing collaborative performance and efficiency in mixed-motive environments such as the Prisoner's Dilemma or resource sharing tasks.

**Strengths:**

The core advantage of the USE method lies in its ability to overcome the fundamental flaws of goal conflict and inaccurate credit assignment in mixed-motive multi-agent collaboration. Traditional weighted-sum approaches merely perform an external trade-off between two independent objectives, often leading policies to converge to suboptimal extremes of either pure selfishness or altruism. In contrast, USE proposes a more elegant solution: through structured decomposition and unified individual rewards (for instance, decomposing individual rewards into dependent and independent components), it establishes an intrinsic linkage between self-interest and collective contribution within the reward signal itself. This intrinsic coupling ensures that while agents maximize their newly defined utility, their policy gradients are automatically aligned with the collective optimum, thereby accurately crediting behaviors that contribute incrementally to team outcomes. Ultimately, this mechanism enables USE to identify a Pareto-improving equilibrium strategy, which in experiments demonstrates both higher collective efficiency (total return) and sufficient preservation of individual interests, significantly enhancing collaborative performance in mixed-motive environments.

**Weaknesses:**

W1: In mixed-motive scenarios, the assumption of being able to access the private rewards (or utility functions) of other agents significantly limits the applicability of many existing multi-agent algorithms in real-world settings.

W2: The performance difference of the algorithm across different seeds is too large; the algorithm's stability seems to be lacking.

W3: https://anonymous.4open.science/r/QPC-B6FD The link is invalid/empty.

**Questions:**

Q1: Equations 1 and 2 mention the necessity of using the global state. This is inappropriate under the assumption of limited agent observation (i.e., a Partially Observable environment), as each agent can only acquire its local observation and should not be able to perceive all environmental information beyond its sensory range, even if the information regarding other agents is excluded. If this is indeed the case (i.e., using the global state is required), then your problem definition should be reformulated as a Markov Game with Global Observation. Furthermore, for the experimental comparison, were the baseline methods you adopted also permitted to access global or non-local environmental information extending beyond their local observation space during the observation phase?

Q2: Regarding MAPPO's significantly weaker fairness performance compared to USE in PD and CG environments, what is the primary reason for this disparity? Additionally, can you provide the training curves for individual rewards of MAPPO and USE in PD and CG?Whether all compared methods (including baselines) utilize the environment's raw reward for fairness evaluation?

Q3: Can you provide a visual comparison of the Q-value accuracy and stability achieved by USE versus baseline algorithms?

Q4: Can you provide a heat map illustrating the change in the cooperation rate of the USE algorithm under different parameter combinations of the Prisoner's Dilemma (PD) game?

Q5: Your method's training is based purely on a self-play training paradigm. If your opponent were to be switched to another algorithm, how would its performance be? Would USE be exploitable? For instance, if the opponent were an Always Defect strategy or an RL-driven one like PPO, a simple verification experiment could be performed in the Prisoner's Dilemma (PD)

---

> ### Author Response · Authors · 2025-11-21
>
> **Q1:** https://anonymous.4open.science/r/QPC-B6FD The link is invalid/empty.
>
> **R1:** Thank you for your review. The complete source code is available in the src folder of the provided repository, which was last updated on June 9, 2025. It is possible that the code was not fully visible initially due to network latency or because the src directory was collapsed by default (requiring a manual click to expand it). We kindly suggest revisiting the link to access the full code content.
>
> **Q2:** Issue of accessing the private rewards and global states: Accessing other agents' private rewards limits real-world applicability in mixed-motive settings. Furthermore, Equations 1 and 2 rely on the global state, which is limited in the Partially Observable setting, where agents should only access local observations. If the global state is required, the problem should be redefined as a Markov Game with Global Observation. Finally, did the baseline methods also have access to global or non-local information during execution?
>
> **R2:** Thank you for your question. We clarify that our method operates without accessing either the private rewards of other agents or the global state during the execution. Instead, it strictly relies on the agent's local observation, as formulated in Equation 14,
> $$\nabla_{\theta_i} J(\theta_i)
> = \mathbb{E}\big[\nabla_{\theta_i} \log \pi_i(a_t^i \mid o_t^i, \tau_t^i)\, (A_t^i + \lambda g_t^i)\big],$$ where the input to the policy network consists solely of the local observation $o_t^i$ and individual history trajectory (encoded by the RNN hidden state, summarizing the sequence $\tau_t^i = (o_0^i, a_0^i, r_0^i, \dots, o_{t-1}^i, a_{t-1}^i, r_{t-1}^i, o_t^i$)). Regarding Equations 1 and 2, we utilize environmental information to compute independent values only during the training phase. While the environmental information is formally pruned from the global state in our equation, in real-world applications, environmental information is usually easier to obtain than the global state.
>
> Formally, USE aligns with the Centralized Training with Decentralized Execution (CTDE) framework. Currently, the majority of multi-agent methods, such as QMIX [1] and MAPPO [2], follow this framework as well. In our experiments, the baseline methods, such as MAPPO, AGA, and others, also adopt the same framework, where they can access non-local environmental information during training, but during execution, they rely entirely on local observations.
>
> [1] Rashid, Tabish, et al. "Monotonic value function factorisation for deep multi-agent reinforcement learning." Journal of Machine Learning Research 21.178 (2020): 1-51.
>
> [2] Yu, Chao, et al. "The surprising effectiveness of PPO in cooperative multi-agent games." Advances in neural information processing systems 35 (2022): 24611-24624.
>
> **Q3:** The performance difference of the algorithm across different seeds is too large; the algorithm's stability seems to be lacking.
>
> **R3:** Thank you for your comment. We would like to clarify that the standard deviation of our algorithm is indeed comparable to the baselines, and we will support this from two aspects:
>
> First of all, as shown in Figures 4 and 5 of the paper, the standard deviation of USE is comparable to the baselines.
> Second, to provide a more intuitive comparison, we present the following table 1 with the performance and standard deviation of USE and the baseline methods in the Cleanup, Harvest, and Allelopathic-Harvest environments. We observe that when compared to methods with suboptimal performance (e.g., PPO and MAPPO in the Cleanup environment), USE indeed exhibits a larger standard deviation. However, this is primarily because USE achieves a performance magnitude 100–1000 times higher than these baselines. This performance gap naturally leads to a larger scale of absolute fluctuations, which is reflected in the standard deviation metric.
>
> To further validate this issue, we computed the Coefficient of Variation (CV, standard deviation divided by mean), as shown in Table 2. It can be observed that, when taking the model's own performance variability into account, the CV of USE is comparable to that of most baseline methods.
>
> |Table 1|USE|AGA|SLI|MOA|IAI|MAPPO|PPO|
> |-|:-:|:-:|:-:|:-:|:-:|:-:|:-:|
> |Cleanup |**1238.3 ± 102.5**| 327.4 ± 83.3 | 708.1 ± 142.3 | 607.3 ± 166.8 | 543.6 ± 181.3 | 0.7 ± 2.0 | 11.8 ± 12.8 |
> |Harvest |**1096.0 ± 87.8**| 796.7 ± 87.4 | 329.2 ± 54.2  | 231.4 ± 60.4  | 213.0 ± 50.4 | 349.8 ± 74.6 | 224.3 ± 17.5 |
> |Allelopathic-Harvest|**94.5 ± 10.4**| 33.8 ± 33.3 | 72.7 ± 11.4 | 37.8 ± 16.6 | 21.7 ± 15.7 | -18.8 ± 29.6 | 56.1 ± 20.7 |
>
> |Table 2 | USE   |  AGA  |  SLI  |  MOA  |  IAI  |  MAPPO  |  PPO  |
> | -| :-:  | :-:  | :-:  | :-:  | :-:  | :-:  | :-:  |
> | Cleanup        |**0.083**| 0.254 | 0.201 | 0.275 | 0.334 | 2.857 | 1.085 |
> | Harvest        | 0.080| 0.110 | 0.165  | 0.261  | 0.237 | 0.213 |**0.078**|
> | Allelopathic-Harvest |**0.110**| 0.986 | 0.157 | 0.439 | 0.724 | -1.575 | 0.369 |

---

> ### Author Response · Authors · 2025-11-21
>
> **Q4:** Can you provide a visual comparison of the Q-value accuracy and stability achieved by USE versus baseline algorithms?
>
> **R4:** To evaluate the Q-value accuracy and stability achieved by USE, we compare it against MAPPO and PPO, since both baselines use standard critic architectures that represent typical implementations. Specifically, Appendix B.5 (Figure 11) reports the mean and standard deviation of the critic loss across 5 random seeds: the mean critic loss reflects the accuracy of Q-value predictions, while the standard deviation reflects their stability. Figure 11(a) shows the absolute critic loss for the three algorithms, and Figure 11(b) shows the relative loss, defined as critic loss divided by the mean absolute Q-value (critic loss / |Q-values|).
>
> As shown in Figure 11(a) and 11(b), USE exhibits convergence behavior and stability comparable to MAPPO and PPO. The results from both absolute and relative loss metrics confirm that USE maintains high Q-value estimation accuracy and stability throughout the training process.
>
> **Q5:** Regarding MAPPO's significantly weaker fairness performance compared to USE in PD and CG environments, what is the primary reason for this disparity? Additionally, can you provide the training curves for individual rewards of MAPPO and USE in PD and CG?Whether all compared methods (including baselines) utilize the environment's raw reward for fairness evaluation?
>
> **R5:** The primary reason for MAPPO's lower fairness lies in its purely cooperative optimization paradigm. MAPPO treats the global/collective objective as the individual objective for all agents and directly optimizes the shared collective return. However, in mixed-motive games, the collective objective is not perfectly aligned with each agent's individual objective. As a result, MAPPO can converge to solutions where some agents are systematically sacrificed or exploited for the sake of maximizing the global return, which leads to poorer fairness.
>
> In contrast, USE achieves fairness by unifying individual and collective rewards. Instead of solely maximizing the global sum, USE optimizes the individual Q-value coupled with the collective objective via the Value Link ($g_t^i$). This mechanism ensures that agents promote collective cooperation while simultaneously safeguarding their individual rewards, driving the policy toward a balanced equilibrium (e.g., mutual cooperation) rather than unilateral exploitation.
>
> Following your suggestion, we have added the training curves of the individual rewards for MAPPO and USE in PD and CG in Appendix B.6.1 (Figure 12). From these curves, we observe that the disparity between the two agents' rewards under MAPPO can be quite large, especially in the Chicken Game (CG).
>
> Regarding your question "whether all compared methods (including baselines) utilize the environment's raw reward for fairness evaluation", we confirm that all compared methods (including baselines) use the raw individual rewards from the environment for fairness evaluation.
>
> **Q6:** Can you provide a heat map illustrating the change in the cooperation rate of the USE algorithm under different parameter combinations of the Prisoner's Dilemma (PD) game?
>
> **R6:** We thank the reviewer for this suggestion. To provide a comprehensive visualization of the algorithm's robustness under varying incentives, we conducted a sensitivity analysis on the payoff matrix.
>
> Specifically, we fixed the Reward ($R=10$) and Punishment ($P=5$), and systematically varied the two key parameters that define the intensity of the dilemma:
> - Temptation ($T \in$ \{13, 15, 17\} ): Representing the incentive to exploit a cooperator (Greed).
> - Sucker's Payoff ($S \in$ \{-2, 0, 2\}): Representing the cost of being exploited (Risk).
>
> The heat map shows that : (1) USE achieves a very high cooperation rate across all nine variants, indicating that USE robustly promotes cooperation under the tested PD parameter combinations. (2) The cooperation rate is lower in the variant with T = 17 and S = 2. This is because the (C, D) outcome grants a large temptation payoff to the defector (T = 17), while the cooperator's payoff in that outcome (S = 2) makes the total payoff for (C, D) equal to T + S = 19, which is nearly the same as the mutual-cooperation total payoff 2R = 20. Such a payoff configuration substantially increases the incentive to defect; nevertheless, the cooperation rate in that heat-map cell remains high (>0.96). This provides evidence of the method's robustness to changes in the underlying PD payoff parameters.
>
> |Table 3|T=13|T=15|T=17|
> |:-:|:-:|:-:|:-:|
> |**S=-2**|\[[(10,10), (-2,13)], [(13,-2), (5,5)]]|\[[(10,10), (-2,15)], [(15,-2), (5,5)]]|\[[(10,10), (-2,17)], [(17,-2), (5,5)]]|
> |**S=0**|\[[(10,10), (0,13)], [(13,0), (5,5)]]|\[[(10,10), (0,15)], [(15,0), (5,5)]]|\[[(10,10), (0,17)], [(17,0), (5,5)]]|
> |**S=2**|\[[(10,10), (2,13)], [(13,2), (5,5)]]|\[[(10,10), (2,15)], [(15,2), (5,5)]]|\[[(10,10), (2,17)], [(17,2), (5,5)]]|

---

> ### Author Response · Authors · 2025-11-21
>
> **Q7:** Your method's training is based purely on a self-play training paradigm. If your opponent were to be switched to another algorithm, how would its performance be? Would USE be exploitable? For instance, if the opponent were an Always Defect strategy or an RL-driven one like PPO, a simple verification experiment could be performed in the Prisoner's Dilemma (PD).
>
> **R7:** We appreciate the reviewer's insightful and thoughtful question. To evaluate whether USE is exploitable, we conducted a simple verification experiment in the Iterated Prisoner's Dilemma where USE played 1,000 rounds against an Always Defect opponent. We recorded the number of cooperative and defective actions taken by USE and computed the cooperation rate (cooperative actions / total actions). The results are summarized in the table below. Against the pure-defection opponent, USE's cooperation rate was 10.2%, indicating that USE largely refrains from cooperating when faced with a consistently selfish opponent and therefore is not substantially exploited in this scenario.
>
> | Table 4 | Number of cooperation | Number of defections | Cooperation rate |
> | --------   | :-----:  | :----:  | :----:  |
> | value        | 102 | 898 | 10.2% |

---

> > ### Author Response · Authors · 2025-11-25
> > **Follow-up: Do Our Responses Resolve Your Main Concerns?**
> >
> > Dear Reviewer 8Ba6,
> >
> > We appreciate your time and effort in reviewing our paper. We have provided detailed responses to your insightful comments and concerns, which we hope have sufficiently addressed the points raised. Some of the key clarifications include the following:
> >
> > 1. **Code Access:** We clarified that the full source code is available in the repository's src folder.
> >
> > 2. **Private Rewards and Global State:** We explained that our method operates solely on local observations during execution, without access to private rewards or the global state.
> >
> > 3. **Performance Stability:** We provided additional statistics, including the Coefficient of Variation, to show that USE's performance variability is comparable to baseline methods.
> >
> > We sincerely hope that these clarifications have resolved your concerns. If they have, we kindly request that you consider adjusting your evaluation score to reflect these updates. We highly value your feedback, as it continues to help us improve the quality of our work.
> >
> > Should you have any further questions or comments, please feel free to share them with us. We look forward to your further feedback.
> >
> > Best regards,
> >
> > The Authors

---

> > > ### Comment · Reviewer_8Ba6 · 2025-11-26
> > > **Thanks for your response, some follow-up questions**
> > >
> > > Thanks for your response. Some of my concerns have been resolved. I have some follow-up questions.
> > >
> > > 1. In response to Q2, the weakness I was referring to is that during the training phase, the method requires access to other agents' private information—specifically, their rewards and observations—which remains impractical for real-world mixed-motive scenarios. However, your reply addressed the execution phase, where agents rely only on local observations. While that is a valuable point, it does not resolve the concern raised in Weakness 1 regarding the training requirements.
> > >
> > > 2. Regarding the response to Q3, my point about Q-value accuracy concerns whether it genuinely reflects how an agent's action benefits the collective, rather than referring to the training loss itself.
> > > To give a concrete example, in the Cleanup environment, cleaning waste may offer an individual an immediate reward of zero, yet it is crucial for the group's long-term benefit. In algorithms like A2C or PPO, the Q-value for taking such an action in a given state tends to be underestimated. In your method, when faced with similar dilemmas, how do the Q-values from the converged network evaluate different actions—particularly those with delayed collective benefits? Could you explain why the Q-values produced by your approach in these cases are more accurate or informative than those from other baselines?
> > > Furthermore, taking the Harvest environment as another example, where does the key difference in Q-value estimation lie between your method and the baselines? What specific mechanism leads to your method's superior performance? It would be very helpful if you could illustrate this with concrete instances or empirical observations, rather than general analysis.
> > >
> > > 3. Thank you for your response to Q6. You mentioned that the (C, D) outcome yields a total payoff of T + S = 19, which is very close to the mutual-cooperation total of 2R = 20. Given that mutual cooperation offers a slightly higher collective return, could you help me understand why your method did not converge toward that outcome?
> > >
> > > I have a few follow-up questions that might help clarify this:
> > >
> > > a) During your experiments, did you apply any normalization to the rewards—for instance, scaling them to a range like (0,1)—while preserving their relative differences? If so, could this have influenced the agent's perception of the payoffs?
> > >
> > > b) In the training process, does your method directly optimize the advantage function?
> > >
> > > c) How did MAPPO perform under the same payoff configuration? Since MAPPO is designed to optimize collective returns, and mutual cooperation clearly offers a higher collective payoff here, one might expect it to consistently converge to full cooperation. If that is the case, would that mean MAPPO outperforms USE in this specific scenario?
> > > If so, this might highlight a meaningful limitation of your method in certain mixed-motive settings, and I would appreciate your perspective on this observation.

---

> > > > ### Author Response · Authors · 2025-11-27
> > > > **Further response to the question**
> > > >
> > > > **Q8:** In response to Q2, the weakness I was referring to is that during the training phase, the method requires access to other agents' private information—specifically, their rewards and observations—which remains impractical for real-world mixed-motive scenarios. However, your reply addressed the execution phase, where agents rely only on local observations. While that is a valuable point, it does not resolve the concern raised in Weakness 1 regarding the training requirements.
> > > >
> > > > **R8:** We sincerely appreciate your clarification.
> > > >
> > > > 1. We do use agents' private information during training. When aggregating the global value, the value function $Q_t^{\text{tot}}$​ combines the $Q_t^i$ from all agents, using the global reward as a constraint. This aggregation process does indeed involve accessing all agents' information.
> > > >
> > > >     *In Q2, you mentioned that Equations 1 and 2 rely on the global state. Here, you may have misunderstood that our USE accesses the full global state (including other agents' observations and private rewards), whereas in fact we only use the pruned state ($s_t^i$) as input. In real-world scenarios, environmental information is often easily accessible. For example, in a mixed-motive cooperation setting where multiple users share a computing cluster, the current CPU load (environmental information) is broadcast in real-time to all agents.*
> > > >
> > > > 2. It is worth noting that most mixed-motive cooperation methods published in top-tier conferences also access other agents' private information to varying degrees during training. Works following the CTDE framework, such as SLI[1], AGA[2], LIO[3], and DIAL[4], use the global state $s_t$ during the training. Works following the DTDE framework (despite claiming decentralized training and execution), such as IAI[5], MOA[6] and LASE[7], require access to agents' private rewards or real-time actions. Therefore, in mixed-motive cooperation, it is common for methods to access other agents' information during training, and it is not unique to USE.
> > > >
> > > > 3. We agree that eliminating the use of other agents' information is a crucial issue in mixed-motive cooperation, and this is a common challenge in multi-agent reinforcement learning. However, this issue is beyond the scope of this paper, which focuses on addressing the issue of policies converging to overly selfish or altruistic behaviors in mixed-motive settings. We will leave it to future research.
> > > >
> > > > [1] Roesch, et al. "Selfishness level induces cooperation in sequential social dilemmas." Proc. of the 23rd International Conference on Autonomous Agents and Multiagent Systems (AAMAS 2024). International Foundation for Autonomous Agents and Multiagent Systems (IFAAMAS), 2024.
> > > >
> > > > [2] Li, et al. "Aligning individual and collective objectives in multi-agent cooperation." Advances in Neural Information Processing Systems 37 (2024): 44735-44760.
> > > >
> > > > [3] Yang, et al. "Learning to incentivize other learning agents." Advances in Neural Information Processing Systems 33 (2020): 15208-15219.
> > > >
> > > > [4] Lin, et al. "Information design in multi-agent reinforcement learning." Advances in Neural Information Processing Systems 36 (2023): 25584-25597.
> > > >
> > > > [5] Li, et al. "Aligning individual and collective objectives in multi-agent cooperation." Advances in Neural Information Processing Systems 37 (2024): 44735-44760.
> > > >
> > > > [6] Jaques, et al. "Social influence as intrinsic motivation for multi-agent deep reinforcement learning." International conference on machine learning. PMLR, 2019.
> > > >
> > > > [7] Kong, et al. "Learning to balance altruism and self-interest based on empathy in mixed-motive games." Advances in Neural Information Processing Systems 37 (2024): 135819-135842.

---

> > > > > ### Author Response · Authors · 2025-11-27
> > > > > **Further response to the question**
> > > > >
> > > > > **Q9:** Regarding the response to Q3, my point about Q-value accuracy concerns whether it genuinely reflects how an agent's action benefits the collective, rather than referring to the training loss itself. To give a concrete example, in the Cleanup environment, cleaning waste may offer an individual an immediate reward of zero, yet it is crucial for the group's long-term benefit. In algorithms like A2C or PPO, the Q-value for taking such an action in a given state tends to be underestimated. In your method, when faced with similar dilemmas, how do the Q-values from the converged network evaluate different actions—particularly those with delayed collective benefits? Could you explain why the Q-values produced by your approach in these cases are more accurate or informative than those from other baselines? Furthermore, taking the Harvest environment as another example, where does the key difference in Q-value estimation lie between your method and the baselines? What specific mechanism leads to your method's superior performance? It would be very helpful if you could illustrate this with concrete instances or empirical observations, rather than general analysis.
> > > > >
> > > > > **R9:** Thank you for this helpful clarification regarding Q-value accuracy. In Figure 8 of the paper, we already visualize how a converged USE network evaluates different actions in the Cleanup environment. To directly address your concern, we additionally compare the converged Q-values of USE and PPO in both Cleanup and Harvest, as shown in the new comparison presented in Appendix B.10, Figure 14.
> > > > >
> > > > > In Cleanup, we observe that: (1) PPO clearly underestimates the value of the Clean action, which is crucial for the collective. Cleaning encourages the apple tree to produce apples, but the produced apples are often harvested by other agents, making it difficult for the cleaning agent to receive individual rewards. As a result, PPO assigns a Q-value of zero to the Clean action, discouraging the agent from cleaning and ultimately harming the long-term collective reward. In contrast, USE's dependent module captures the positive interaction value of cleaning, assigning a substantially higher Q-value to cleaning behavior, thereby encouraging cooperative actions. (2) PPO assigns relatively high Q-values to the Fire action, as defecting by shooting others enables an agent to harvest more quickly and increase its individual return. However, this leads to the undermining of collective cooperation. In contrast, USE's dependent module assigns a negative interaction value to Fire, reflecting its detrimental impact on the collective and discouraging such behavior.
> > > > >
> > > > > In Harvest, we observe a similar pattern: PPO tends to overestimate the value of Fire, whereas USE's dependent module assigns negative values to this action, capturing its detrimental effect on the collective and leading to superior group performance.
> > > > >
> > > > > These empirical comparisons illustrate that, in terms of your proposed criterion regarding whether Q-values genuinely reflect how an agent's action benefits or harms the collective, the Q-values produced by USE are more informative than those of PPO.

---

> > > > > > ### Author Response · Authors · 2025-11-27
> > > > > > **Further response to the question**
> > > > > >
> > > > > > **Q10:** Thank you for your response to Q6. You mentioned that the (C, D) outcome yields a total payoff of T + S = 19, which is very close to the mutual-cooperation total of 2R = 20. Given that mutual cooperation offers a slightly higher collective return, could you help me understand why your method did not converge toward that outcome?
> > > > > >
> > > > > > **R10:** Thank you very much for this question. We would first like to clarify that, under this payoff configuration, USE in fact learns an almost fully cooperative outcome. In our experiments, the average cooperation rate reaches about 0.975 (notably, the cooperation rate of MAPPO is 0.977), meaning that in the majority of cases both agents choose to cooperate.
> > > > > >
> > > > > > As discussed in Appendix B.7, one factor contributing to this imperfect result (a cooperation rate of 0.975 instead of 1) is that the difference between the total payoff under mutual cooperation and unilateral defection is very small in this extreme setting: 2R=20 is only one unit higher than T+S=19. This makes the guidance provided by the VLink signal $g_t^i$ relatively weak. Consequently, the individual advantage component can occasionally dominate and still assign a slightly positive advantage to defection, especially under function approximation and finite-sample noise.
> > > > > >
> > > > > >
> > > > > >
> > > > > > **Q(a):** During your experiments, did you apply any normalization to the rewards—for instance, scaling them to a range like (0,1)—while preserving their relative differences? If so, could this have influenced the agent's perception of the payoffs?
> > > > > >
> > > > > > **R(a):** In our implementation, we do not apply any normalization to the rewards themselves. The only normalization step we use is applied to the advantage values, which are normalized before being used in the policy gradient update.
> > > > > >
> > > > > >
> > > > > >
> > > > > > **Q(b):** In the training process, does your method directly optimize the advantage function?
> > > > > >
> > > > > > **R(b):** Thank you for helping us clarify this point. Our actor is indeed based on an advantage-style policy gradient, but the optimization target is not the individual advantage alone. Concretely, as defined in Eqs. (13)–(14), we first compute the normalized individual advantage $A_t^i$. We then combine the normalized advantage with the VLink signal $g_t^i$, which measures the marginal contribution of the dependent Q-value to the collective Q-value. The policy gradient update takes the form
> > > > > > $$ \nabla_{\theta^i} J(\theta^i) = \nabla_{\theta^i} \log \pi^i(a_t^i|o_{t}^i,\tau_t^i) \cdot \left[A^i_t + \lambda g_t^i \right]. $$
> > > > > > Thus, USE directly optimizes a unified objective that balances the individual advantage $A_t^i$ and the collective-effect term $g_t^i$, rather than the individual advantage function alone.
> > > > > >
> > > > > >
> > > > > >
> > > > > > **Q(c):** How did MAPPO perform under the same payoff configuration? Since MAPPO is designed to optimize collective returns, and mutual cooperation clearly offers a higher collective payoff here, one might expect it to consistently converge to full cooperation. If that is the case, would that mean MAPPO outperforms USE in this specific scenario? If so, this might highlight a meaningful limitation of your method in certain mixed-motive settings, and I would appreciate your perspective on this observation.
> > > > > >
> > > > > > **R(c):** Thank you for raising this important point. To address your question, we trained MAPPO under the same payoff configuration where T + S = 19, and we report the results in the table below.
> > > > > >
> > > > > > | |USE|MAPPO|
> > > > > > |-|:-:|:-:|
> > > > > > |Cooperation Rate|0.975|0.977|
> > > > > >
> > > > > > We summarize our observations as follows:
> > > > > > (1) Although MAPPO is designed to maximize the collective return and indeed learns a highly cooperative policy, its average total payoff oscillates near the ideal value of 20 rather than converging to exactly 20. Note that in the implementation of MAPPO, we strictly adhere to the official settings.
> > > > > > (2) Under this extreme payoff configuration with a strong temptation to defect (T = 17) and only a slight collective disadvantage for (C, D) (T + S = 19 vs. 2R = 20), USE attains a collective performance comparable to MAPPO, with a cooperation rate of 0.975, just slightly below MAPPO's cooperation rate of 0.977. It attains a total return that is very close to that of MAPPO.
> > > > > >
> > > > > > These observations indicate that even in high-temptation settings where unilateral defection is individually attractive yet not extremely harmful to the collective outcome, USE still exhibits near-perfect cooperation, and MAPPO does not clearly outperform USE in terms of collective return. Although MAPPO performs marginally better than USE in this task, the difference is negligible. Importantly, in more complex mixed-motive environments, USE achieves higher overall performance and significantly better fairness.

---

### Author Response · Authors · 2025-11-21
**General response**

We want to express our sincere gratitude to all the reviewers for their valuable feedback and insightful comments. We also appreciate the reviewers' recognition of the strengths of our paper, which they all have highlighted: **novelty**, **strong** performance, and **extensive/detailed experiments and ablations**.

We have refined the manuscript by adding more details and experiments to improve the depth of our analysis. Specifically:

**a:** We implemented and evaluated a variant, local-USE, which removes the dependency on the pruned state ($s_t^i$) even during training. This demonstrates the scalability and applicability of our method under restricted information settings (detailed in Appendix B.9).

**b:** We provided a deeper qualitative analysis of the learned strategies (e.g., dynamic role switching) and included training curves of individual rewards to explicitly illustrate the fairness advantage of USE (detailed in Appendix B.6).

**c:** We expanded our robustness evaluations, including sensitivity analysis on payoff parameters (Appendix B.7) and exploitability tests against defecting opponents (Appendix B.8), to validate the method's reliability in complex scenarios.

**d:** We revised the theoretical analysis to formally guarantee the stability of our algorithm under neural function approximation (via projection operators and experience replay), addressing the theoretical concern about unbounded gradients (presented in Appendix C.1).

**e:** We added a comprehensive discussion on the game-theoretical perspective, clarifying the fundamental differences between USE and existing cooperative game theory methods (e.g., CIAO), in Appendix C.2.

---

### Author Response · Authors · 2025-12-03
**Final Summarization of Author-Reviewer Discussion**

Dear Area Chair,

We would like to express our sincere gratitude for your valuable contributions to the community during this challenging period. Below, we provide a brief summary of our rebuttal.
### **Summary of Our Paper**
Our paper introduces USE for mixed-motive cooperation, where agents must balance collective welfare and individual returns. Purely self-interested behavior can degrade the collective outcome and even hurt long-term individual payoffs, while focusing only on the collective leaves cooperative agents vulnerable to exploitation. To address this issue, USE decomposes an agent's individual Q-value into an independent component, driven by its own behavior, and a dependent component, shaped by interactions with others. It further introduces the Value Link (VLink), which captures how the dependent component influences the collective Q-value, thereby coupling individual and collective objectives. This coupling allows policies that improve individual returns to simultaneously promote collective welfare, mitigating overly selfish or overly altruistic behaviors in mixed-motive settings.
### **Overview of the Discussion**
Our paper was reviewed by three reviewers. Before the rollback, h3Yf raised the score to **8**, recognizing the significance of our method for mixed-motive cooperation. 8Ba6 acknowledged that most concerns had been addressed and engaged in a productive second-round discussion. VQH3 did not provide further feedback, but we responded point by point and believe we have sufficiently addressed their concerns.
### **Strengths**
The reviewers identified the following strengths of our paper:
- USE decomposes each agent's individual value function into independent and dependent components, providing an elegant and novel approach that represents a meaningful theoretical advance `8Ba6, h3Yf`.
- USE effectively balances individual gain and collective welfare, achieving high collective efficiency without sacrificing individual returns `8Ba6, h3Yf, VQH3`.
- USE includes informative ablations and extensive experiments across diverse environments and baselines, systematically evaluating performance, generalization, and fairness `h3Yf, VQH3`.
- The paper is clearly written and well structured `h3Yf, VQH3`.
- The paper presents a rigorous analysis of convergence and boundedness `h3Yf`.

### **Raised Concerns and Our Responses**
We summarize the primary concerns raised by the reviewers, along with our corresponding responses:
1. **Reviewers 8Ba6 and VQH3 were concerned about whether CTDE reduces the scalability of USE.**
- **Information requirements:** We clarified that mixed-motive cooperation is still a cooperative multi-agent task, so using a CTDE framework for training is conceptually reasonable. Prior work on mixed-motive cooperation likewise relies, to varying degrees, on other agents' information during training, and our assumption is consistent with existing methods.
- **Local-USE:** We relaxed the input constraints of USE and carried out experiments in Appendix B.9 and found that it still outperforms the baselines, indicating that USE scales well under weaker information assumptions.
2. **Reviewers 8Ba6 and h3Yf were concerned about USE’s critic accuracy and actor stability.**
- **Critic Accuracy:** We compared critic losses in Appendix B.5 and the converged Q-values of USE in Appendix B.10, showing that USE provides accurate and semantically meaningful Q-value estimates for actions.
- **Actor Stability:** We formally analyzed the method based on global gradient clipping and experience replay in Appendix C.1, ensuring the stability of the actors.
3. **Reviewer 8Ba6 was concerned about USE's robust cooperative behavior.**
- **Fairness:** We included training curves in Appendix B.6, where USE maintains fairness with lower inequality.
- **Cooperation:** We conducted a payoff-matrix sensitivity analysis in Appendix B.7, showing that USE still achieves near-perfect cooperation even in extreme settings.
- **Exploitability:** We reported that USE's cooperation rate is only 10.2% when facing an Always Defect opponent in Appendix B.8, indicating that it effectively avoids long-term exploitation.
4. **Reviewer h3Yf wonders how the hyperparameter $\lambda$ is selected.**
- **Hyperparameter $\lambda$**: We analyzed the choice of $\lambda$ in Appendix B.4. We found that USE performs best when the magnitudes of the individual advantage term and the VLink term are balanced.
5. **Reviewer VQH3 was concerned about the choice of hypernetwork inputs.**
- **Hypernetwork Input:** We analyzed the use of local observations in the hypernetwork in Appendix B.1.3, showing that it yields the best performance compared to using global or pruned states.

We sincerely appreciate the time and effort the AC has dedicated to maintaining a fair and balanced review process. Given the broad set of additional analyses we have conducted in this rebuttal, we kindly ask the AC to review our revised paper.

---

### Meta-Review · Area_Chair_2Cg5 · 2026-01-06

**Summary:**

This paper studies mixed-motive cooperation in multi-agent reinforcement learning. Instead of optimizing a weighted sum of individual and collective rewards, it decomposes each agent’s objective into an independent component and an interaction-dependent component, and introduces the Value Link (VLink) to couple the dependent component with the collective value. This design aims to align individual optimization with collective outcomes. Reviewers generally agree that the method is technically novel and well motivated.

**Reviewer Concerns:**

A central concern raised by multiple reviewers is that the method requires access to other agents’ private rewards or information during training, which substantially limits its applicability in realistic mixed-motive settings. Although the authors clarified that execution is decentralized and provided additional experiments and analysis, they ultimately acknowledge that such information is indeed required during training and position this limitation as outside the scope of the paper.

**Reviewer Scores:**

While one reviewer became fully positive after the rebuttal, at least one initially negative reviewer maintained their core objection regarding the information assumptions during training, explicitly noting that this issue remains unresolved. Although the authors provided careful justifications and empirical support for other aspects of the method, the fundamental concern about the practicality of the training requirements was not eliminated.

---

### Decision · Program_Chairs · 2026-01-26

Reject